# Phase-field modeling and machine learning of electric-thermal-mechanical breakdown of polymer-based dielectrics

Zhong-Hui Shen[1,5], Jian-Jun Wang[2,5], Jian-Yong Jiang[1], Sharon X. Huang[3], Yuan-Hua Lin[1], Ce-Wen Nan[1], Long-Qing Chen[2] & Yang Shen[1,4]

Understanding the breakdown mechanisms of polymer-based dielectrics is critical to achieving high-density energy storage. Here a comprehensive phase-field model is developed to investigate the electric, thermal, and mechanical effects in the breakdown process of polymer-based dielectrics. High-throughput simulations are performed for the P(VDF-HFP)-based nanocomposites filled with nanoparticles of different properties. Machine learning is conducted on the database from the high-throughput simulations to produce an analytical expression for the breakdown strength, which is verified by targeted experimental measurements and can be used to semiquantitatively predict the breakdown strength of the P(VDF-HFP)-based nanocomposites. The present work provides fundamental insights to the breakdown mechanisms of polymer nanocomposite dielectrics and establishes a powerful theoretical framework of materials design for optimizing their breakdown strength and thus maximizing their energy storage by screening suitable nanofillers. It can potentially be extended to optimize the performances of other types of materials such as thermoelectrics and solid electrolytes.

[1] School of Materials Science and Engineering, State Key Lab of New Ceramics and Fine Processing, Tsinghua University, 100084 Beijing, China. [2] Department of Materials Science and Engineering, The Pennsylvania State University, University Park, PA 16802, USA. [3] Information Sciences and Technology, The Pennsylvania State University, University Park, PA 16802, USA. [4] Center for Flexible Electronics Technology, Tsinghua University, 100084 Beijing, China. [5]These authors contributed equally: Zhong-Hui Shen, Jian-Jun Wang. Correspondence and requests for materials should be addressed to J.-J.W. (email: wjj8384@gmail.com) or to L.-Q.C. (email: lqc3@psu.edu) or to Y.S. (email: shyang_mse@tsinghua.edu.cn)

Polymer-based dielectrics are the most promising material candidates for high-density energy storage applications due to their high breakdown strength, low dielectric loss, high direct-current resistivity, and flexibility[1–6]. While increasing the loading electric field can effectively enhance the amount of energy stored in a dielectric, the probability of breaking down the dielectrics catastrophically increases as well. Despite tremendous efforts have been invested to understand the dielectric breakdown mechanism[7–10], the dielectric breakdown is still among the least understood physical phenomena due to the complex electrical, thermal, mechanical, and chemical interactions within a composite dielectric.

A number of possible dielectric breakdown mechanisms have been proposed, including electrical, thermal, mechanical, and partial discharge breakdown. The electric breakdown typically represents the irreversible damage due to the high electric field-induced avalanche multiplication of free charge carriers and hence the electric current when carriers acquire sufficient kinetic energy between collisions with the matrix to give a high probability of ionization, with generation of more carriers, and without suffering sufficient recombination[11–13]. The thermal breakdown may take place when the heat generated from Joule heating of the dielectric cannot be effectively dissipated to the surroundings[4,11,14]. The mechanical effect on the breakdown stems from the increase in the elastic energy density due to the electric field-induced shrinkage of the insulation thickness, which depends on the Young's modulus, dielectric constant, and the applied electric field[11,15,16]. The partial discharge results from the high electric field-induced gas ionization in the voids embedded in a polymer dielectric as the local electric field is intensified at the void/polymer interfaces because the voids filled with gas have a lower dielectric constant than the polymer matrix[13,17,18]. The coexistence and coupling of these mechanisms make the breakdown process extremely complicated and unpredictable, particularly in material systems with complex microstructures such as polymer/ceramic nanocomposites[2,6,7,10,19].

In this work, we develop a comprehensive phase-field model of dielectric breakdown by incorporating the electrical, thermal, and mechanical effects. It is based on a preliminary phase-field model for studying the electrostatic and electrothermal breakdown processes in polymer-based nanocomposites[20,21]. Using the poly (vinylidene fluoride-hexafluoropropylene) (P(VDF-HFP)) as an example, the electrical, thermal, and mechanical effects on its breakdown strength are analyzed to unveil the underlying physical mechanisms at different temperatures and applied electric fields. Furthermore, by parameterizing the dielectric constant, electrical conductivity, and Young's modulus of the nanofillers to represent different filler materials, high-throughput phase-field simulations are performed to obtain the corresponding breakdown strengths of the P(VDF-HFP)-based nanocomposites. Then, machine learning is performed on the high-throughput simulation results to produce an analytical expression for the breakdown strength as a function of the dielectric constant, electrical conductivity, and Young's modulus of the nanofillers. Additional phase-field simulations and targeted experiments are performed to validate the analytical function, which can then be used to make extremely quick prediction of the breakdown strength of P(VDF-HFP)-based nanocomposites with potential nanofillers.

## Results

### Phase-field simulations of the breakdown process

Here, we take the pure P(VDF-HFP) polymer as an example to perform a two-dimensional phase-field simulation of the breakdown process under different applied electric fields along $y$ direction ($E_y^{app}$) and temperatures. Three intermediate states of the breakdown process at 295, 323, and 363 K are exhibited in Fig. 1a–i, respectively, assuming that the initial breakdown phase is nucleated from the two needle electrodes. At 295 K, the breakdown phase begins to grow under an applied electric field of ~680 kV mm$^{-1}$ shown in Fig. 1a. The polymer is on the verge of breakdown around ~700 kV mm$^{-1}$. When the temperature is increased to 323 K, the electric field threshold of breakdown drops to ~565 kV mm$^{-1}$, and the polymer is almost entirely broken down at ~580 kV mm$^{-1}$. If the temperature is further increased to 363 K, the electric field threshold further decreases to ~405 kV mm$^{-1}$ with the near breakdown field decreasing to ~415 kV mm$^{-1}$. Therefore, a moderate temperature increase of 68 K from 295 to 363 K leads to a breakdown strength reduction by 285 kV mm$^{-1}$ from 700 to 415 kV mm$^{-1}$.

Figure 1j–r show the distributions of the local electric energy density $f_{elec}$, Joule heat energy density $f_{Joule}$, and strain energy density $f_{strain}$ corresponding to the final breakdown states in Fig. 1c, f, i, respectively. More energy density distributions are shown in Supplementary Discussion. All energy densities at the front and inside of the breakdown path are much higher than at other regions, resulting in the forward continual growth of the breakdown path. For the electric energy density shown in Fig. 1j, m, p, it is less dependent on the temperature because the dielectric constant changes only slightly from 295 K to 363 K. The differences in the electric energy density mainly arise from the different magnitudes of the applied electric field according to $f_{elec} = \frac{1}{2} \varepsilon_0 \varepsilon_r E^2$.

However, at these three temperatures, the thermal energy densities from Joule heating $f_{Joule}$ are completely different, as shown in Fig. 1k, n, q. Due to the high dependence on temperature, the electrical conductivity of P(VDF-HFP) varies by several orders of magnitude from ~10$^{-10}$ S m$^{-1}$ at 295 K to ~10$^{-8}$ S m$^{-1}$ at 363 K. Therefore, $f_{Joule}$ is the highest at 363 K among the three temperatures, although the breakdown strength 415 kV mm$^{-1}$ at 363 K is only 59% of the value 700 kV mm$^{-1}$ at 295 K. Figure 1l, o, r show the strain energy density distributions, corresponding to the final breakdown states in Fig. 1c, f, i, respectively. The strain energy density increases with the electric field, but decreases with the Young's modulus. At 363 K, the breakdown electric field and Young's modulus are 415 kV mm$^{-1}$ and 214 MPa, respectively, giving rise to a strain energy density ~3.14×10$^5$ J m$^{-3}$, which has a similar order of magnitude to ~4.37×10$^5$ J m$^{-3}$ at 295 K calculated with a breakdown strength of 700 kV mm$^{-1}$ and a Young's modulus of 982 MPa (see Supplementary Table 1). Therefore, the strain energy densities displayed in Fig. 1l, o, r show no clear difference, although the breakdown strength and Young's modulus at these temperatures are different.

Figure 2a shows the breakdown strength as a function of temperature obtained from experimental measurements and predictions from the Stark–Garton model[16,22] and the phase-field model incorporating different breakdown mechanisms. The Stark–Garton model describes the pure electromechanical breakdown for polymers, and it predicts a much higher breakdown strength than the experimental measurement. This means there must exist other mechanisms in the breakdown process of P(VDF-HFP) polymer. The phase-field model predicts different breakdown strength versus temperature features, depending on the mechanisms that are incorporated into the model. When only incorporating electric energy in the phase-field model, the predicted breakdown strength $E_b^{elec}$ is larger than experimental measurements at temperatures higher than 295 K. Incorporating both electric and Joule heat energies in the phase-field model,

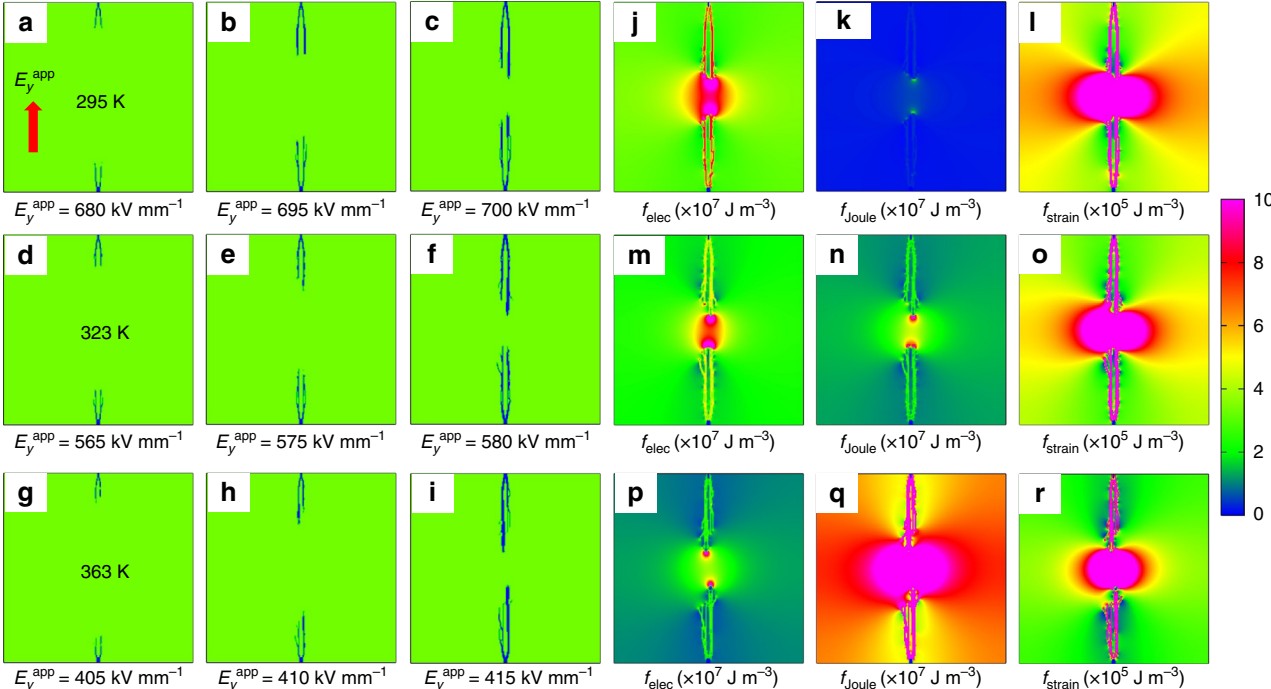

**Fig. 1** Breakdown evolution and corresponding energy density distributions. The breakdown evolution processes of poly(vinylidene fluoride-hexafluoropropylene) (P(VDF-HFP)) polymer predicted by the extended phase-field model at different applied electric fields and temperatures of **a–c** 295 K, **d–f** 323 K, and **g–i** 363 K. The applied field $E_y^{app}$ is along the $y$ direction and is shown by the red arrow in **a**. The initial breakdown phase is nucleated from the two needle electrodes. **j–r** show the electric energy density $f_{elec}$, Joule heat energy density $f_{Joule}$, and strain energy density $f_{strain}$ for the breakdown states of **c**, **f**, and **i**

the predicted breakdown strengths ($E_b^{elec+Joule}$) agree with experimental measurements when the temperature is lower than 340 K. When incorporating electric, Joule heat, and strain energies, the phase-field model can predict breakdown strengths ($E_b^{elec+Joule+strain}$) that are very consistent with experimental measurements in the specified temperature range (295–363 K). Therefore, at 295 K, the electric effect is sufficient to describe the breakdown process. With temperature increasing, the electrical–thermal coupling has to be taken into account to describe the breakdown versus temperature features. When the temperature is above 343 K, the electrical–thermal–mechanical coupling is necessary to describe the breakdown behavior as a function of temperature.

Figure 2b shows separately the average electric, Joule heat, and strain energy densities as functions of the applied electric field based on simulation results at 295, 323, and 363 K using the phase-field model. All energy densities increase with temperature. When a breakdown occurs, all energy densities show a jump. At 295 and 323 K, the electric energy density is higher than the Joule heat and strain energy densities, verifying that the electric effect dominates the breakdown process at these two temperatures. However, at 363 K, the average Joule heat energy density becomes the highest and dominates the breakdown process. In combination with electric and strain energies, an electrical–thermal–mechanical coupling is established as the breakdown mechanism at high temperatures.

**Breakdown mechanisms in various polymer dielectrics**. To identify the breakdown mechanism for an arbitrary polymer-based dielectric, the dependences of electric, Joule heat, and strain energy densities on the applied electric field and material parameters, including the dielectric constant, electrical conductivity, and the Young's modulus, are mapped using phase-field simulations. Figure 3a shows the mapped electric energy density as a function of the applied electric field (0–10³ kV mm⁻¹)

and the dielectric constant (1–10⁴), which suggests that higher dielectric constant and higher electric field result in higher electric energy, as expected. At an electric field of $10^3$ kV mm⁻¹ and a dielectric constant of $10^4$, the electric energy density can reach ~$4.43 \times 10^{10}$ J m⁻³, as shown at the top right corner of Fig. 3a. Similarly, Fig. 3b suggests that higher electrical conductivity and higher electric field also result in higher Joule heat energy density. At an electric field of $10^3$ kV mm⁻¹ and an electrical conductivity of $10^{-4}$ S m⁻¹, the Joule heat energy density can be as high as ~$10^{12}$ J m⁻³.

While the strain energy density is affected by both the dielectric constant and the Young's modulus, their effects, in combination with the applied electric field, on the strain energy density are separately calculated. Figure 3c shows the mapped strain energy density as a function of the applied electric field and the dielectric constant at a fixed Young's modulus of 1 GPa. Higher electric field and higher dielectric constant give rise to a higher strain energy density. At an electric field of $10^3$ kV mm⁻¹ and a dielectric constant of $10^4$, the strain energy density is ~$10^{11}$ J m⁻³. Figure 3d shows the dependence of the strain energy density on the applied electric field and the Young's modulus at a fixed dielectric constant of 100. In contrast to the dielectric constant, higher Young's modulus leads to lower strain energy density. At an electric field of $10^3$ kV mm⁻¹ and a Young's modulus of $10^6$ Pa, the strain energy density is ~$10^{10}$ J m⁻³.

By summarizing the calculations shown in Fig. 3a–d, we can draw schematic diagrams to understand the breakdown behavior and identify the possible breakdown mechanism for different polymer-based dielectrics. Figure 4a phenomenologically shows the variation of the energy profile for polymers under physical stimuli. Curve 1 describes a double-well energy density as function of the order parameter $\eta$, and the energy barrier height between $\eta = 0$ (unbroken phase) and $\eta = 1$ (broken phase) represents how difficult a local point can be broken down. With

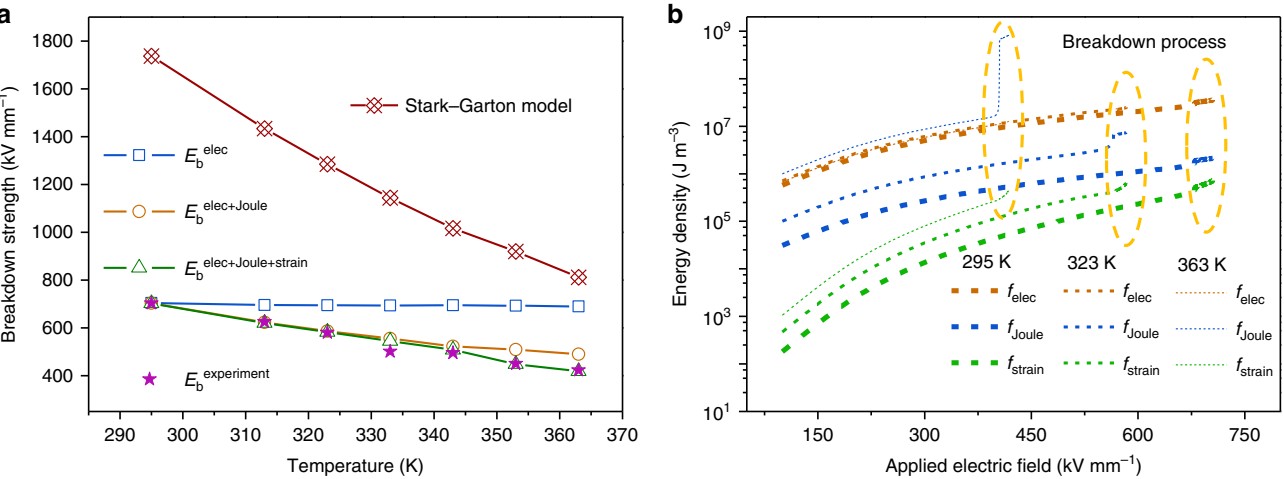

**Fig. 2** Breakdown strength and energy densities of poly(vinylidene fluoride-hexafluoropropylene) (P(VDF-HFP)) polymer. **a** The breakdown strengths[22] as functions of temperature obtained from experimental measurements and predictions from the Stark–Garton model[16] and the phase-field model. $E_b^{elec}$, $E_b^{elec+Joule}$, and $E_b^{elec+Joule+strain}$ denote the breakdown strengths predicted by the phase-field model considering only electric energy, the combination of electric and Joule heat energy, and the combination of electric, Joule heat, and the strain energy, respectively. **b** Comparisons of the average electric energy density, Joule heat energy density, and the strain energy density at different applied electric fields and temperatures. The dotted circles show the abrupt increase of energy density during the breakdown process

**Fig. 3** High-throughput calculations of energy densities. Mapping results of **a** the electric energy density as a function of the dielectric constant and the applied electric field, **b** Joule heat energy density as a function of the dielectric constant and the applied electric field, **c** strain energy density as a function of the electrical conductivity and the applied electric field, and **d** strain energy density as a function of the Young's modulus and the applied electric field

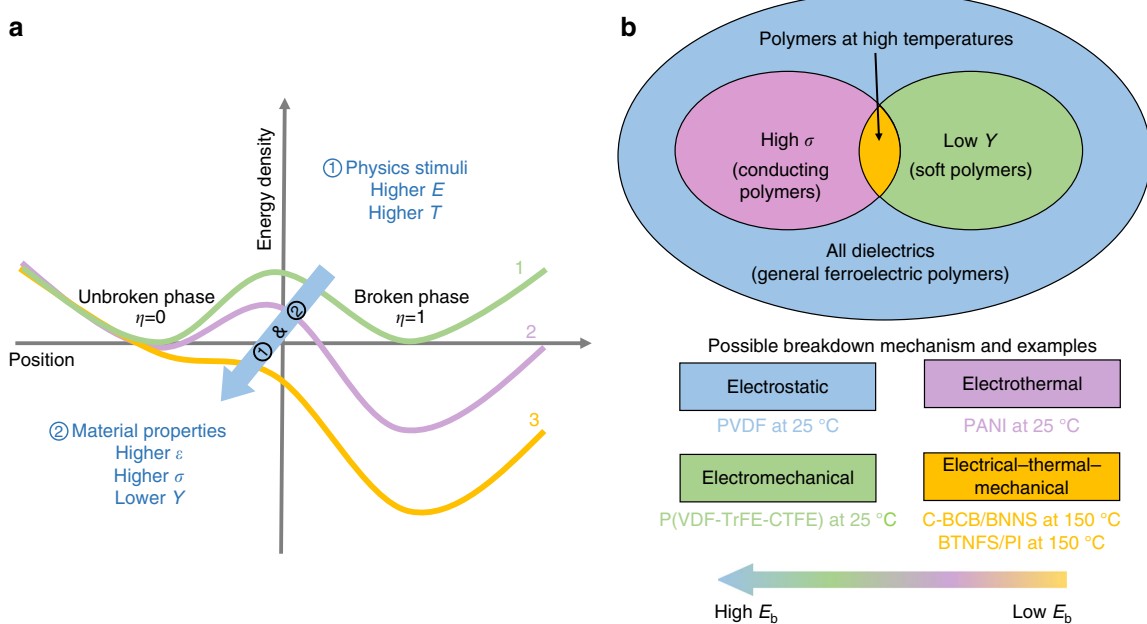

**Fig. 4** Schematic diagrams for understanding and classifying breakdown mechanisms. **a** Variation of the energy profile as function of the order parameter for polymers under physical stimuli. Each solid line denotes the double-well energy profile under different stimuli. The blue arrow signifies that the energy profile can be tilted by physics stimuli or changes of material properties. **b** Identification of breakdown mechanisms in polymer-based material systems. Examples of poly(vinylidene fluoride-hexafluoropropylene) (PVDF), poly(vinylidenefluoride-trifluoroethylene-chlorotrifluoroethylene) (P(VDF-TrFE-CTFE)), polyaniline (PANI), cross-linked divinyltetramethyldisiloxanebis(benzocyclobutene) BNNS composite (C-BCB/BNNS), and BaTiO₃ nanofiber/polyimide (BTNFS/PI) are from experiments[23–27]. Different color regions represent possible breakdown mechanisms of different dielectrics. From the electrostatic breakdown (in blue) to electrical–thermal–mechanical breakdown (in yellow), the breakdown strength decreases

the increase of external physical stimuli such as the electric field $E$ or temperature $T$ or both, the energy barrier drops and energy profile tilts, leading to a lower energy state at $\eta = 1$ than that at $\eta = 0$, as illustrated by the variation from curve 1 to curve 3 shown in Fig. 4a. Once the physical stimuli is sufficiently large, the energy barrier vanishes and the breakdown occurs. The height of this energy barrier is also related to the material parameters, including the dielectric constant $\varepsilon$, the electrical conductivity $\sigma$ and the Young's Modulus $Y$. For a material with higher dielectric constant $\varepsilon$, higher electrical conductivity $\sigma$ and lower Young's Modulus $Y$, the energy barrier is lower. Thus, the breakdown strength will be lower.

Figure 4b shows the respective breakdown mechanisms and corresponding representative examples. For general ferroelectric polymers at room temperature, for example, PVDF[23], the electric breakdown is the dominant mechanism. While for conducting polymers such as polyaniline[24] at room temperature, the breakdown process may be dominated by electrothermal mechanism due to the high electrical conductivity. For soft polymers with high dielectric constant and ultralow Young's modulus (usually ≤500 MPa) such as P(VDF-TrFE-CTFE) (poly(vinylidenefluoride-trifluoroethylene-chlorotrifluoroethylene))[25] at room temperature, the electromechanical effect is likely an important factor for the breakdown process. However, as the temperature increases to near the glass transition temperature $T_g$ or the melting temperature $T_m$ of the polymer, the electrical conductivity increases, and the Young's modulus decreases dramatically, which will eventually lead to an electrical–thermal–mechanical breakdown. This mechanism is likely to occur in most polymer-based dielectrics, for example, C-BCB/BNNS (cross-linked divinyltetramethyldisiloxanebis(benzocyclobutene) BNNS composite)[26] at 150 °C and BTNFS/PI[27] at 150 °C, which are targeted for high temperature applications, as shown in Fig. 4b. Therefore, the possible breakdown mechanism of

dielectrics can be qualitatively identified from the material parameters of the dielectrics, which is useful to the design of polymer-based nanocomposites for enhancing their breakdown strength. For example, if the electrothermal breakdown mechanism is identified for a polymer, nanofillers with low electrical conductivity and high thermal conductivity can be filled in the polymer to mitigate the Joule heat energy. This guideline coincides with the design principle of polymer nanocomposites with high breakdown strength proposed in previous experiments[26].

**High-throughput phase-field simulations and machine learning**. There have been intensive experimental activities to synthesize polymer dielectrics with enhanced breakdown strength[3,7,28–31], most of which largely employed the trial-and-error and intuition-driven approaches to optimizing the microstructures of polymer-based dielectrics. A simple but predictive model, for example, an analytical expression for the breakdown strength as a function of material parameters, can be a guide to the experimental design and synthesis of polymer nanocomposites[32,33], thus shortening the time and reducing the cost for the development of new materials as mandated by the Materials Genome Initiative. Here, by performing high-throughput phase-field simulations and machine learning, we produce an analytical expression for the breakdown strength of P(VDF-HFP)-based nanocomposites.

We assume that the fillers have random distribution and sphere shapes, fixed at 5 vol.%. Figure 5 shows the strategy we propose to employ machine learning to construct the analytical expression. High-throughput simulations are first performed to provide a breakdown strength database for the machine learning. Three properties of the nanofillers, including the dielectric constant $\varepsilon$, electrical conductivity $\sigma$, and Young's modulus $Y$,

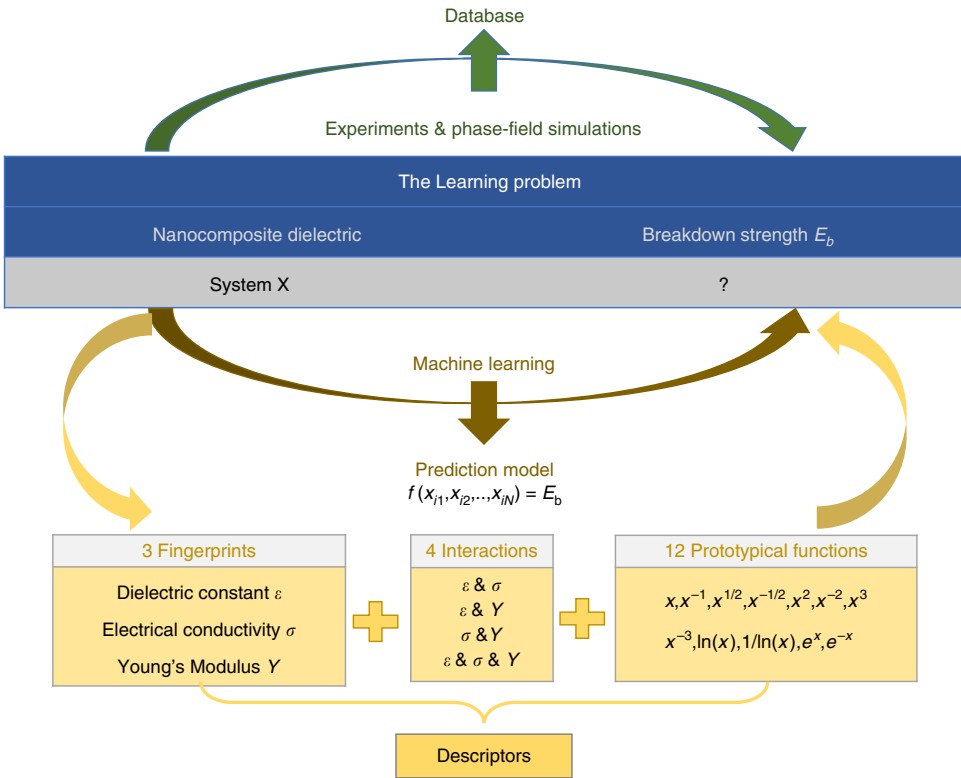

**Fig. 5** Machine learning strategy. Schematic workflow of the machine learning strategy for producing an analytical expression for the breakdown strength of poly(vinylidene fluoride-hexafluoropropylene) (P(VDF-HFP))-based nanocomposites from the breakdown strength database generated by the high-throughput phase-field simulations. In the machine learning process, 3 fingerprints, 4 interactions, and 12 prototypical functions are employed to perform least-squares regressions and screenings

are chosen as variables for the high-throughput simulations and fingerprints for the machine learning. As shown in Fig. 5, 12 prototypical functions and four interactions are considered to generate descriptors to perform the regression analysis by least-squares regression (LSR)[34–36]. The coefficient of determination ($R^2$) of the LSR is used as the criterion for screening of the descriptors. As described in Supplementary Fig. 1, three rounds of LSR and screening are performed, which eventually led to a predictive expression for the breakdown strength as a function of the three primary properties.

Figure 6a displays breakdown strengths calculated from the high-throughput phase-field simulations for the P(VDF-HFP) nanocomposites filled by 5 vol.% nanofillers with different dielectric constant, electrical conductivity, and Young's modulus, which are also used as the training dataset for the machine learning. The breakdown strength has a stronger dependence on the electrical conductivity than the dielectric constant and the Young's modulus. More specifically, with the electrical conductivity, the most sensitive material parameter, increasing from $2.12 \times 10^{-13}$ to $2.12 \times 10^{-7}$ S m$^{-1}$ while maintaining the dielectric constant at 13.5 and the Young's modulus at 0.982 GPa, the normalized breakdown strength decreases from about 1.34 to 0.32. However, with the Young's modulus, the least sensitive material parameter, increasing from 0.982 MPa to 982 GPa while maintaining the dielectric constant at 13.5 and the electrical conductivity at $2.12 \times 10^{-10}$ S cm$^{-1}$, the normalized breakdown strength just increases from 0.42 to 1.04.

Then, we perform machine learning on Fig. 6a. After the first and second rounds of screening of LSR results, an expression without considering the interactions among the fingerprints

is obtained,

$$y = 0.9058 - 0.01175 \ln x_1 - 0.06104 \ln x_2 - 0.0164 x_3^{-1/2}, \quad (1)$$

which has a coefficient of determination $R^2 = 0.8899$ (see Supplementary Tables 2, 3, 4). Here, $y$, $x_1$, $x_2$, and $x_3$ represent $E_b^{\text{composite}}/E_b^{\text{matrix}}$, $\varepsilon_{\text{filler}}/\varepsilon_{\text{matrix}}$, $\sigma_{\text{filler}}/\sigma_{\text{matrix}}$, and $Y_{\text{filler}}/Y_{\text{matrix}}$, respectively. When considering interactions among the fingerprints, an updated expression is obtained after the third round of screening, which gives a higher coefficient of determination $R^2 = 0.9099$ (Supplementary Table 5).

However, taking into account the need for a simple and practical analytical expression, we choose a simpler expression considering only the interaction between $\sigma$ and $Y$, that is,

$$\frac{E_b^{\text{composite}}}{E_b^{\text{matrix}}} = 0.9058 - 0.01175 \ln \frac{\varepsilon_{\text{filler}}}{\varepsilon_{\text{matrix}}} - 0.06767 \ln \frac{\sigma_{\text{filler}}}{\sigma_{\text{matrix}}} - 0.01640 \left(\frac{Y_{\text{filler}}}{Y_{\text{matrix}}}\right)^{-1/2} + 0.001 \left(\frac{Y_{\text{filler}}}{Y_{\text{matrix}}}\right)^{-1/2} \ln \frac{\sigma_{\text{filler}}}{\sigma_{\text{matrix}}}. \quad (2)$$

This expression gives a coefficient of determination value $R^2 = 0.9093$ as the predictive function. More details of the regression analysis can be found in Supplementary Methods. Eq. (2) suggests that $E_b^{\text{composite}}$ decreases with $\varepsilon_{\text{filler}}$ and $\sigma_{\text{filler}}$, but increases with $Y_{\text{filler}}$. It implies that the breakdown strength of the polymer nanocomposite can be enhanced by nanofillers with higher Young's modulus but lower dielectric constant and electrical conductivity. In general, the Young's modulus of the ceramic nanofillers is much larger than that of the polymer matrix; therefore, seeking for nanofillers with lower dielectric constant and lower electrical conductivity will be more critical to improve the breakdown strength of nanocomposites.

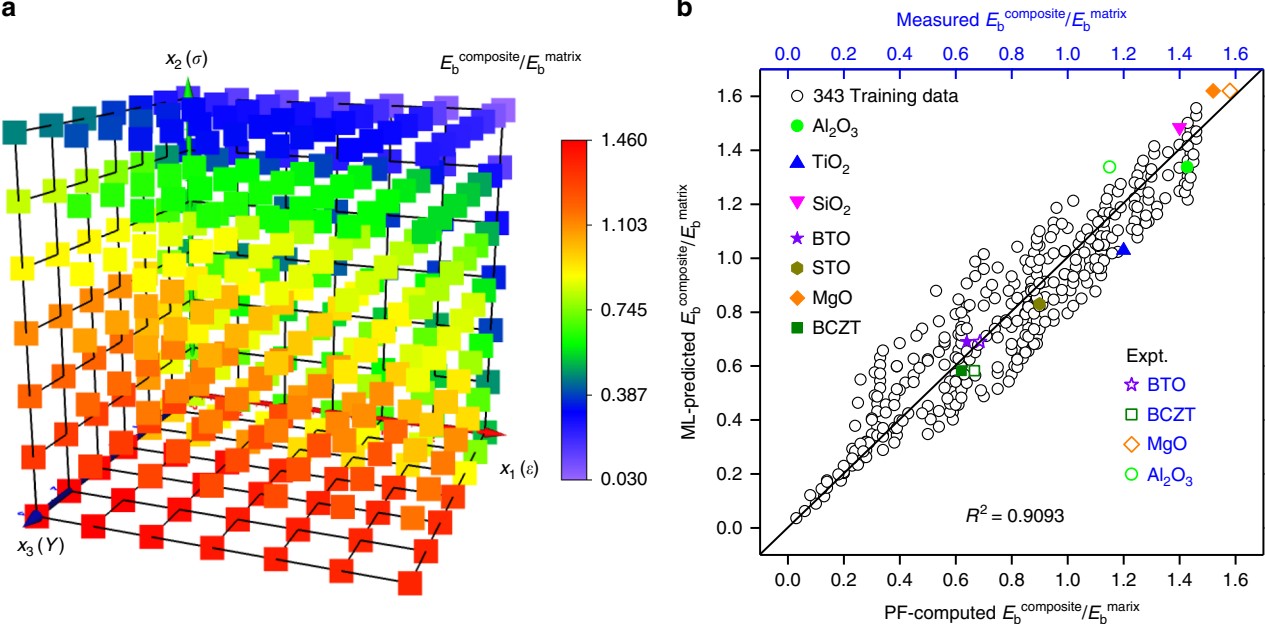

**Fig. 6** High-throughput phase-field simulation and machine learning results. **a** High-throughput simulation results of the breakdown strength for poly (vinylidene fluoride-hexafluoropropylene) (P(VDF-HFP)) nanocomposites filled by 5 vol% nanofillers with different dielectric constant, electrical conductivity, and Young's modulus. Here, $x_1(\varepsilon)$, $x_2(\sigma)$, and $x_3(Y)$ represent $\varepsilon_{filler}/\varepsilon_{matrix}$, $\sigma_{filler}/\sigma_{matrix}$, and $Y_{filler}/Y_{matrix}$, respectively. **b** Comparisons of breakdown strengths between the phase-field model and the machine learning prediction, verified by seven new types of fillers as the testing set (not included in the training set, marked as colored solid symbols) and some experimental results (colored open symbols). The experimental measurement for the breakdown strength of $Al_2O_3$/P(VDF-HFP) nanocomposite is performed in this work. The experimental breakdown strengths for BTO/P(VDF-HFP), BCZT/P(VDF), and MgO/P(VDF) nanocomposites are from the literatures[37–39]

Moreover, the absolute value of the factor before the term $\ln(\sigma_{filler}/\sigma_{matrix})$ in the expression is larger than other factors before the other two terms, indicating that the breakdown strength of the P(VDF-HFP)-based nanocomposites is more sensitive to electrical conductivity than to dielectric constant and Young's modulus, which is consistent with the calculations shown in Fig. 6a. Note that Eq.(2) can automatically incorporate the temperature effect on the breakdown strength as long as the temperature dependencies of dielectric constant, electrical conductivity, and Young's modulus are available and incorporated. Aside from the LSR, we also tried another machine learning method, the back-propagation neural network (BPNN) with details described in Supplementary Discussion. In comparison to LSR, the BPNN exhibits better prediction ability of the breakdown strength. However, the BPNN cannot give an expression of breakdown strength as functions of the dielectric constant, electrical conductivity, and Young's modulus, thereby it is less convenient for experimental researchers to make a quick estimation of the breakdown strength for a new material system.

The reliability of the produced analytical expression for breakdown strength from the machine learning is tested by comparing the predictions with additional phase-field simulations and available experimental measurements. As shown in Fig. 6b, there are seven data points marked with colored solid symbols which are not included in the training dataset of Fig. 6a. They represent filler materials of $Al_2O_3$, $TiO_2$, $SiO_2$, $BaTiO_3$, $SrTiO_3$, MgO, and $Ba_yCa_{1-y}(Zr_xTi_{1-x})O_3$ (BCZT), respectively, whose material parameters used in the phase-field simulation and machine learning prediction are listed in Supplementary Table 5. These seven symbols are dispersed around the solid line which represents $E_b^{phase-field} = E_b^{regression-analysis}$, indicating that the breakdown strengths for these seven nanocomposites calculated

from phase-field simulations agree with the predictions from the machine learning, thereby validating the expression in Eq. (2).

The predicted breakdown strengths using Eq. (2) for BTO/P (VDF-HFP), BCZT/P(VDF-HFP), and MgO/P(VDF-HFP) nanocomposites are also compared with existing congeneric experimental measurements[37–39] (colored open symbols on Fig. 6b). These three open symbols are close to the solid line which also represents $E_b^{measurement} = E_b^{regression-analysis}$, indicating that the predicted breakdown strengths for these three polymer nanocomposites agree with the experimental measurements. To further test the reliability of the breakdown strength function in Eq. (2), a $Al_2O_3$/P(VDF-HFP) nanocomposite filled with 5 vol.% $Al_2O_3$ nanoparticles is fabricated and characterized in this work (Supplementary Fig. 8). The $E_b^{composite}/E_b^{matrix}$ from the experimental measurement is about 1.15, which agrees fairly well with 1.34 predicted from Eq. (2), further validating the breakdown strength function produced from the machine learning. Many factors such as voids, space charge effects[40–42] and incomplete crystallization[11,13,43,44] of the polymer that are not incorporated into the phase-field model may cause this difference between the phase-field-based machine learning and the experimental results. For example, the existence of voids may cause partial discharge at the void/polymer interfaces where the local electric field is intensified. A high concentration of space charges can cause the increases in the electrical conductivity. The crystallinity of polymers may also affect the breakdown strength, arising from the different transport behaviors of charge carriers in the amorphous phase and crystalline phase. Unfortunately, the introduction of a large number of nanofillers can easily introduce these effects due to the incompatibility of ceramics nanofillers and polymers. Therefore, the preparation of high-quality polymer nanocomposites is extremely important to achieve a high breakdown strength.

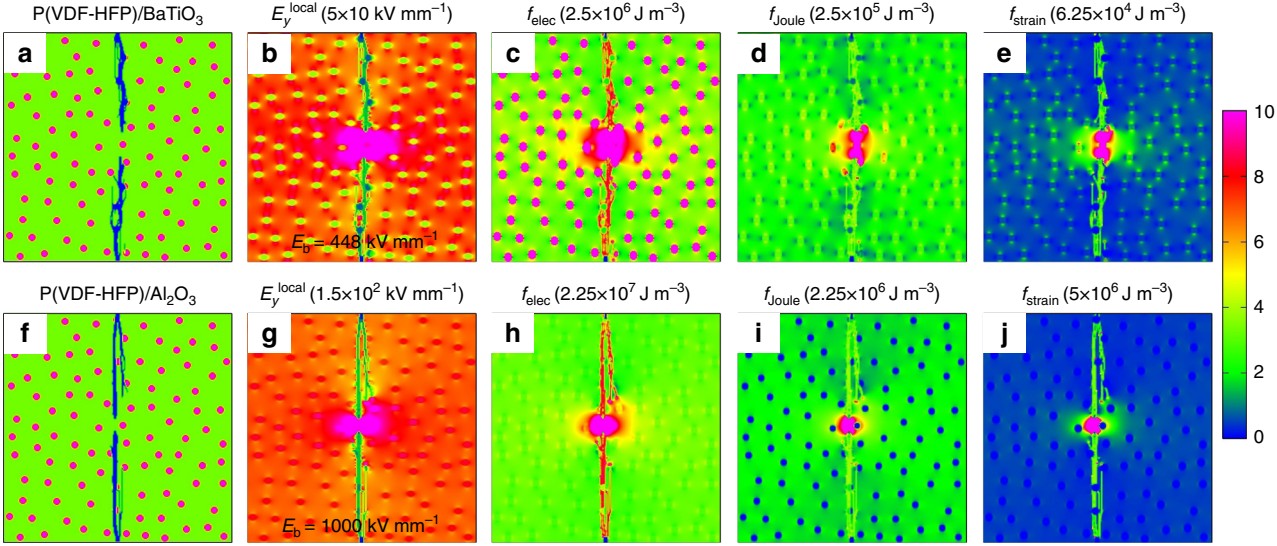

**Fig. 7** Breakdown morphologies of polymer nanocomposites from phase-field simulations. The final state of the breakdown process of **a** BaTiO$_3$/P(VDF-HFP) nanocomposite and (**f**) Al$_2$O$_3$/P(VDF-HFP) nanocomposite with corresponding electric-field distributions shown in **b**, **g**, electric energy density distributions shown in **c**, **h**, Joule heat energy density distributions shown in **d**, **i**, and strain energy density distributions shown in **e**, **j**

## Discussion

The presented phase-field model can be used to investigate the electrical, thermal, and mechanical effects on the breakdown process of polymer-based dielectrics. It reveals that the electric effect is much more dominant than the thermal and mechanical effects on the breakdown process in the pure P(VDF-HFP) polymer at room temperature, resulting in an electric breakdown at ~700 kV mm$^{-1}$. With temperature rising, the electrical conductivity increases and the Young's modulus decreases, eventually leading to an electrical–thermal–mechanical breakdown at a much lower breakdown field.

By filling nanoparticles into the polymer, the breakdown strength can be altered, depending on the material parameters of the filler nanoparticles. Employing the high-throughput phase-field simulations and machine learning, the filler effects on the breakdown strength of the polymer nanocomposite can be readily predicted. For example, by filling 5 vol.% BaTiO$_3$ and Al$_2$O$_3$ into P(VDF-HFP), the predicted breakdown strength at room temperature will be decreased by ~36% to 448 kV mm$^{-1}$ and increased by ~43% to 1000 kV mm$^{-1}$, respectively. The underlying mechanisms causing this difference can be understood from Fig. 7a to j. The electrical conductivity and dielectric constant of BaTiO$_3$ are much larger than P(VDF-HFP), while those of Al$_2$O$_3$ are smaller than the matrix. This leads to the concentration of the local electric field at heterointerfaces in BaTiO$_3$/P(VDF-HFP) (Fig. 7b), while inside the Al$_2$O$_3$ nanofillers in Al$_2$O$_3$/P(VDF-HFP) (Fig. 7g). Due to better insulation, Al$_2$O$_3$ has better endurance capability of electric field than the polymer matrix. Consequently, Al$_2$O$_3$ nanofillers can partake and assimilate the major portion of the voltage in Al$_2$O$_3$/P(VDF-HFP), leading to an enhanced breakdown strength of the nanocomposite. On the other hand, the electric, Joule heat and strain energy density distributions of Al$_2$O$_3$/P(VDF-HFP) shown in Fig. 7h–j are more homogeneous than those of BaTiO$_3$/P(VDF-HFP) shown in Fig. 7c–e. Note that a homogeneous energy distribution is more favorable for avoiding hot spots, and delaying or inhibiting the growth of a breakdown path, thereby benefiting the breakdown strength. It is worth mentioning that the idea to improve the breakdown strength by introducing nanofillers with lower electrical conductivity and dielectric constants into a polymer is consistent with the design principle of polymer nanocomposites filled with hexagonal boron nitride (h-BN) proposed by experimentalists[26,31]. According to the analytical expression and the mechanism analysis above, if nanofillers with low dielectric constant and low electrical conductivity are added into the polymer, the breakdown strength of nanocomposites can be improved.

In summary, a comprehensive electrical–thermal–mechanical breakdown phase-field model is developed to study the breakdown process of polymer-based dielectrics. It reveals a clear breakdown mechanism change of P(VDF-HFP) polymer with temperature, from an electrically dominating breakdown to an electrothermal breakdown, and eventually to an electrical–thermal–mechanical breakdown as temperature increases. By parameterizing the dielectric constant, electrical conductivity, and Young's modulus and analyzing the electric, Joule heat, and strain energy densities at different electric fields, a general principle is established to identify the breakdown mechanism in various polymer dielectrics (Fig. 4). High-throughput phase-field simulations are performed to generate a database on the nanofiller effects on the breakdown strength of P(VDF-HFP)-based nanocomposites, from which machine learning is employed to obtain an analytical expression for the breakdown strength as a function of nanofiller material parameters, including the dielectric constant, electrical conductivity, and Young's modulus. This analytical expression can be employed to readily predict the breakdown strength of polymer nanocomposites filled with nanoparticles with available material parameters. Specific examples include nanocomposites of P(VDF-HFP) filled with oxides such as Al$_2$O$_3$, SiO$_2$, MgO, and TiO$_2$. The machine learning predicts that those nanocomposites should exhibit higher breakdown strength than the pure polymer matrix, and this prediction is verified by both calculations and experiments in this work. The present work can be extended to study the breakdown strength as functions of nanofiller material parameters for polymer nanocomposites filled with other morphologies, such as nanofibers, nanosheets, and nanofillers with arbitrary geometries. This work establishes a theoretical strategy of selecting the best nanofillers to optimize the breakdown performances of polymer-based dielectrics, which will provide guidance to theorists and experimentalists in the design of high-energy-density materials and devices.

## Methods

**Phase-field model of electrical–thermal–mechanical breakdown**. In the phase-field model, a continuous phase-field variable $\eta(\mathbf{r},t)$ is used to describe the temporal

and spatial evolution of the breakdown phase: $\eta(\mathbf{r}, t) = 1$ and $\eta(\mathbf{r}, t) = 0$ denote the broken and unbroken phase, respectively, and the diffuse transitional region represents the interface between the two phases. The total free energy includes the interfacial energy, the electric energy, the Joule heating energy, and the strain energy, that is,

$$F = \int_V \left[ f_{\text{sep}}(\eta(\mathbf{r})) + f_{\text{grad}}(\eta(\mathbf{r})) + f_{\text{elec}}(\mathbf{r}) + f_{\text{Joule}}(\mathbf{r}) + f_{\text{strain}}(\mathbf{r}) \right] dV. \quad (3)$$

A modified Allen–Cahn equation is employed to describe the breakdown phase evolution,

$$\frac{\partial \eta(\mathbf{r}, t)}{\partial t} = -L_0 H(f_{\text{elec}} + f_{\text{Joule}} + f_{\text{strain}} - f_{\text{critical}}) \frac{\delta F}{\delta \eta(\mathbf{r}, t)}, \quad (4)$$

where $L_0$ is the kinetic coefficient related to the interface mobility, $H(f_{\text{elec}} + f_{\text{Joule}} + f_{\text{strain}} - f_{\text{critical}})$ is the Heaviside unit step function, and $f_{\text{critical}}$ is a position-dependent material constant representing the critical energy density of each component in the composite. The purpose of introducing the Heaviside function is to assure that the breakdown phase can grow only if the total energy density ($f_{\text{elec}} + f_{\text{Joule}} + f_{\text{strain}}$) at a local point is greater than its critical energy endurance. In order to obtain the electric-field distribution during microstructure evolution, the spectral iterative perturbation method is employed[45,46]. The detailed procedure for solving Eq. (4) and input parameters can be found in Supplementary Methods.

**Machine learning approach**. Three properties of the filler materials, including the electrical conductivity, dielectric constant, and Young's modulus, are selected as the fingerprints to identify a cause–effect relationship with the breakdown strength. As shown in Fig. 5, candidate expressions and descriptors are generated by cross-multiplying the primary features (i.e., fingerprints) and combining them with the interactions among fingerprints and 12 prototypical function types. Then, the different expression candidates are ranked and screened by their coefficients of determination from Least Squares Regression (LSR), in order to discover the expression giving rise to the highest coefficient of determination.

Here, 343 sets of phase-field simulations using the electrical–thermal–mechanical breakdown model are performed to prepare the training data. After discovering Eq. (2) from the machine learning, seven additional sets of phase-field simulations are performed to validate the produced breakdown strength function, using materials parameters for $Al_2O_3$, $TiO_2$, $SiO_2$, $BaTiO_3$, $SrTiO_3$, MgO, and BCZT. For further validating the machine learning results, the breakdown strengths predicted using Eq. (2) for BTO/P(VDF-HFP), BCZT/P(VDF-HFP), and MgO/P(VDF-HFP) nanocomposites are also compared with existing congeneric experimental measurements (colored open symbols in Fig. 6b) as well as experimental result on $Al_2O_3$/P(VDF-HFP) from this work. More details of the machine learning are shown in Supplementary Discussion.

**Experimental section**. Chemicals were obtained from the following commercial sources and used without further purification: $Al_2O_3$ (China National Chemicals Corporation Ltd.), P(VDF-HFP (Arkema, France, Kynar Flex 2801 with 10 wt.% HFP). P(VDF-HFP) powder was thoroughly dissolved in a mixed solvent of N,N-dimethylformamide and acetone. With the aid of ultrasonic treatment, $Al_2O_3$ nanoparticles were then dispersed into the P(VDF-HFP) solution with a volume fraction 5%. Then, electrospinning and hot pressing process were performed. Finally, a ~15-μm-thick $Al_2O_3$/P(VDF-HFP) nanocomposite was obtained. Thus, the whole fabrication process includes dissolution, ultrasonic treatment, electrospinning, and hot pressing[19,37]. The measured breakdown strengths of the prepared pure P(VDF-HFP) polymer and $Al_2O_3$/P(VDF-HFP) nanocomposites were plotted in a Weibull distribution diagram, which indicates breakdown strengths of 561.2 V mm$^{-1}$ and 647.6 kV mm$^{-1}$ for the pure P(VDF-HFP) polymer and $Al_2O_3$/P(VDF-HFP) nanocomposites, respectively (Supplementary Fig. 8).

## Data availability
The data that support the findings of this study are available from the authors on reasonable request.

## Code availability
The code that supports the findings of this study is available from the authors on reasonable request.

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

## Acknowledgements
Y.S. acknowledge the financial supports by Basic Science Center Program of NSFC (Grant No. 51788104), the NSF of China (Grant Nos. 51625202 and 51572141), the National Key Research and Development Program (Grant No. 2017YFB0701603), and the National Basic Research Program of China (Grant No. 2015CB654603). J.-J.W. and L.-Q.C. acknowledge the partial financial support for this effort from the Donald W. Hamer Foundation for the Hamer Professorship at Penn State.

## Author contributions
Z.-H.S. and J.-J.W. conceived and designed the project. J.-Y.J. prepared and measured the samples. S.X.H. contributed to perform machine learning. Z.-H.S. performed and analyzed the phase-field simulation with help from J.-J.W., Y.-H. L., C.-W. N., L.-Q.C., and Y. S. The project was coordinated by J.-J.W., Y.S., and L.-Q.C., Z.-H.S., and J.-J.W. wrote the manuscript with help from all other authors. All authors contributed to the discussion of results.
