## [Peer Review File · Nature Communications]

Reviewers' comments:

Reviewer #1 (Remarks to the Author):

The manuscript titled "Electric-Thermal-Mechanical Breakdown of Polymer-based Dielectrics: High-Throughput Phase-Field Simulations and Machine Learning" by Shen et al. reports a phase-field model to explore and study the electric, thermal, and mechanical effects in the breakdown process of polymer-based dielectrics. The developed model is applied to carry out high-throughput 2D simulations of breakdown strengths for the P(VDF-HFP)-based nanocomposites filled with nanoparticles of different dielectric elastic and thermal properties. The data is then analyzed within a simple machine learning based framework to produce an analytical expression for the breakdown strength of polymer nanocomposites. Overall the work is carefully carried out and the paper is well written. The work is undoubtedly of high value for those in the community who are working in this specialized field of polymer-dielectric composites for energy storage and related applications. However, in my opinion, neither the research approach nor the insights that come out of the work are novel enough to warrant a publication in Nature Communication.

The phase-field model reported in the present manuscript can be considered an incremental improvement over the recent past works published in the following references: Adv. Mater. 30, 1704380 (2018) and Adv. Energy Mater. 1800509 (2018). The two papers have already reported the details of the electrical and electrothermal phase field simulations. As far as the phase field model development is concerned, the current work can be considered only a minor extension of these works by adding mechanical effects (which seem to play a relatively minor role in dictating the breakdown field strength; looking at Fig. 2 of the manuscript, for instance).

The high throughput framework has also been previously reported in Adv. Mater. 30, 1704380 (2018).

The machine learning analysis and the feature selection approach by considering combinatorial functional forms of a pre-selected set of primary features for mining the analytical relationships such as those presented in the paper have also been reported before [for example, see Chem. Mater. 28, 1304-1311 (2016)]. In fact, the machine learning analysis presented here appears to be highly inspired from the approach and work flow presented in Chem. Mater. 28, 1304-1311 (2016).

Results and findings of the present work do not go any further from confirming the conventional wisdom. Specifically "What new insights have come out as a result of this work?" is not clear.

For the above reasons, the manuscript is more suitable for a specialized journal rather than Nature Communication, which emphasizes on novelty and impact.

Reviewer #2 (Remarks to the Author):

This manuscript presents a continuum, phase-field model for the breakdown of polymer dielectrics, which involves the electric energy, Joule heating, and strain. This model includes significantly more physics than the old Stark-Garton model and is able to reproduce the experimental data well to fairly well.

The authors also use machine-learning techniques on a set of ~400 simulations to arrive at an analytical expression that replicates well the results of the phase-field simulations, including data not included in the training set. The final part of the manuscript deals with predictions and experiments on nanocomposites.

The phase-field model represents a genuine advance over the Stark-Garton expression of breakdown and thus deserves publication in Nature Communications. However, the presentation

and discussion in terms of figures and their descriptions left me wondering about the deeper physical issues and the expected accuracy, which have not been addressed fully. It is also hard to develop deeper physical insight from an examination of a few figures. My suggestions are the following:

(i) The number of experimental data points used to compare to the results of phase-field modeling is rather limited, while the phase-field model uses parameters and approximations. The authors should discuss the extent to which their parameters are uniquely determined and the sensitivity of their results to parameter variations. This is particularly important because the physical properties of complex polymers vary between samples and are not known with high accuracy. What accuracy is expected? Are only the general trends expected to be reproduced or are the modeling results truly quantitative?

(ii) I would like to see a deeper physical description of the model in the main text, perhaps instead of some figures if space is a problem.

(iii) Clearly, the phase-field model uses a continuum description and ignores atomic-scale effects, such as defects and material inhomogeneities. A discussion of these left-out effects would benefit the readers.

Reviewer #3 (Remarks to the Author):

This manuscript developed a comprehensive phase-field model to investigate electric-thermal-mechanical breakdown of polymer-based dielectrics. Until now, both the theoretical and experimental study of the mechanism of breakdown strength of polymer-based composites are rather insufficient to obtain a definite conclusion compared with other dielectric properties. Due to the various influencing factors, such as the electrode system, measuring environment, the shape of the sample, the intrinsic properties of materials and so on, it is unrealistic to depict the breakdown phenomenon with a unified theory. However, it does not affect the contribution of this paper revealing the internal breakdown mechanism of polymer-based composites. Comparison between the calculation and the experiment results indicates that the incorporation of the phase field method is effective. It is of great significant to guide the experiment design with the conclusions from the phase field calculation and the linear regression.

Based on different energies, this model could identify the breakdown mechanisms and predict the breakdown strength of polymer-based dielectrics under different stimulus. Then, high-throughput calculations and machine learning were conducted to produce an analytical expression of breakdown strength. I think this work is interesting and very helpful to design and screen nanocomposites for experiments. I recommend this manuscript to be published after the following comments are addressed.

1. The author mentioned "assuming that the initial breakdown phase is nucleated from the two needle electrodes". Why does the breakdown path start from the electrodes? Please give enough explanation.

2. In this work, the matrix polymer is P(VDF-HFP). Does this model also work in other systems? How about the microstructure of nanocomposites? Does it consider the state of the molecular chain or crystallinity of polymer?

3. For dielectric breakdown, there are so many factors affecting this process. Sometimes, the extrinsic factors, e.g., defects, impurities or air hole, may also be important.

4. Using the analytical expression of dielectric breakdown by machine learning, it has found that fillers like Al₂O₃ or MgO can improve the breakdown strength of nanocomposites. How can we use the expression to find more new materials?

5. The introduce of fillers like Al₂O₃ or MgO can improve the breakdown strength of nanocomposites. But those fillers also lead to the decrease of dielectric constant due to the low intrinsic dielectric constant. So if we want to get high energy density of nanocomposites, how to balance these two parameters?

6. Whether the simulation model is 2D or 3D. if it is 2D, when extending to a three-dimensional condition, the circle representing the ceramic particles will become cylinder rather sphere. It will

deviate from the actual situations and the well agreement between the calculation and the experiment will be less convincing.

7. When simulating the thermal breakdown, whether the heat dissipation is considered? In other works, whether the temperature(363K) is only used to change the temperature-dependent electrical conductivity or used as the ambient temperature?

8. Supporting Information[line601-604]: The kinetic coefficient $L \rightarrow 0$ is not given.

9. Supporting Information[equation 8-9]: Can you give a detailed derivation for eq.8 to eq.9.

10. Supporting Information [equation 11] Please explain the equation more detailed. Where does the expression $(\eta^3 (10-15\eta+6\eta^2))$ come from?

Response to the Reviewers' comments:

Reviewer #1

The manuscript titled "Electric-Thermal-Mechanical Breakdown of Polymer-based Dielectrics: High-Throughput Phase-Field Simulations and Machine Learning" by Shen et al. reports a phase-field model to explore and study the electric, thermal, and mechanical effects in the breakdown process of polymer-based dielectrics. The developed model is applied to carryout high-throughput 2D simulations of breakdown strengths for the P(VDF-HFP)-based nanocomposites filled with nanoparticles of different dielectric elastic and thermal properties. The data is then analyzed within a simple machine learning based framework to produce an analytical expression for the breakdown strength of polymer nanocomposites. Overall the work is carefully carried out and the paper is well written. The work is undoubtably of high value for those in the community who are working in this specialized field of polymer-dielectric composites for energy storage and related applications. However, in my opinion, neither the research approach nor the insights that come out of the work are novel enough to warrant a publication in Nature Communication.

Response: We are thankful to the referee for the careful reading of our manuscript and the positive comments on the value of our work and the writing. We would like to address the concerns by the referee about the novelty and general interest of our work from the following aspects:

- 1) This is the first time that the electric, thermal, and mechanical effects are simultaneously incorporated into a phase-field model of dielectric breakdown, which represents a breakthrough advance in the theory and computation of dielectric breakdown. Such a model is critical for understanding the transition of breakdown mechanisms under different electric fields and temperatures.
- 2) Combining the high-throughput calculations of dielectric breakdown and machine learning in this work is another major advance, making it possible to guide the design of materials to achieve enhanced breakdown strength. For example, based

on the results, we are able to show that some oxides such as Al₂O₃, MgO, SiO₂, and TiO₂ with lower dielectric constant and lower electrical conductivity can be used as nanofillers in polymer nanocomposites to increase the breakdown strength. Such computational guidance is expected to stimulate experimental efforts for validation and further research on high-energy-density polymer nanocomposites.

- 3) The general framework combining high throughput phase-field simulations of responses of microstructures under external stimuli and machine learning can be extended to the understanding and design of other types of materials systems, e.g., optimizing the “ZT” values of a two-phase thermoelectric system with respect to volume fraction, morphology as well as the electric and thermal conductivities of each individual phase. Therefore, this work is of general interest rather than only to the community of polymer composites.

In summary, we believe both of our comprehensive phase-field model of breakdown incorporating thermal, mechanical and electrical effects and the integration of high throughput phase-field simulations and machine learning are novel, and this computational framework can be generally applied to the understanding and design of many other materials systems and is thus of general interest.

The phase-field model reported in the present manuscript can be considered an incremental improvement over the recent past works published in the following references: Adv. Mater. 30, 1704380 (2018) and Adv. Energy Mater. 1800509 (2018). The two papers have already reported the details of the electrical and electrothermal phase field simulations. As far as the phase field model development is concerned, the current work can be considered only a minor extension of these works by adding mechanical effects (which seem to play relatively minor role in dictating the breakdown field strength; looking at Fig. 2 of the manuscript, for instance).

Response: We thank the referee has carefully read our manuscript and previous works.

- 1) It is true that the comprehensive model in this work is built on our previous results. However, the model published in Adv. Mater. 30, 1704380 (2018) only considers the electrical effect, so it can only be used to study the electrical breakdown at room

temperature. Then, we incorporate the thermal effect and published the electrothermal breakdown model in *Adv. Energy Mater.* 1800509 (2018). It can be used to study the breakdown at different temperatures. However, at high temperatures, particularly above the glass transition temperature, the polymer will become softer, and the mechanical effect cannot be neglected. Therefore, incorporating the mechanical effect is required. The model presented in this work is a comprehensive model which can be used to study the breakdown under simultaneous electrical, thermal, and mechanical stimuli and help understand under which breakdown mechanism will dominate under a given set of thermoelectromechanical conditions.

2) In the test example of P(VDF-HFP) in Fig. 2, the mechanical effect does play a relatively minor role in dictating the breakdown field strength, especially when the temperature T is lower than about 342K. However, when $T > 342\text{K}$, the reduction of breakdown strength can reach as high as about 50 kV/mm, which may not be regarded as a minor change for the application of energy storage. In fact, the mechanical effect in this model is strongly related to the material parameters of dielectric constant and especially Young's modulus according to $f_{\text{strain}} = \frac{\epsilon_0^2 \epsilon_r^2 E^4}{8Y}$. It plays more important roles under high electric field and high temperature for soft polymers, like terpolymer with high dielectric constant of about 50 and low Young's modulus of about 200MPa, as verified in left top of Fig. 3(d). Thus, the importance of mechanical effect is material-dependent. Most importantly, the usefulness of this model lies in the fact that one does not have to assume which breakdown mechanism would dominate a priori in predicting the breakdown mechanism under a given condition. Therefore, it can be used to analyze the role of each breakdown mechanism in different systems under different stimulus, which is very helpful for providing guidance for the experimentalists to design materials with high breakdown strength, as summarized in Fig.4.

The high throughput framework has also been previously reported in *Adv. Mater.* 30, 1704380 (2018).

Response: We thank the referee for this question. Yes, we have done high-throughput

calculations in our previous work published in Adv. Mater. 30, 1704380 (2018). However, in previous work, we used the high-throughput calculations to study the microstructure-property relationship: the dependences of effective dielectric constant, breakdown strength, and energy density on the shape and orientation of the nanofillers. In this work, we perform high-throughput phase-field simulations to study the effects of material parameters including the electrical conductivity, dielectric constant, and Young's modulus on the breakdown strength and energy contributions. The previous work emphasizes the microstructure, and this work emphasizes the material parameters of the nanofiller. In addition, the high-throughput simulation results in this work are designed for performing the machine learning to obtain the analytical expression of breakdown strength as function of material parameters.

The machine learning analysis and the feature selection approach by considering combinatorial functional forms of a pre-selected set of primary features for mining the analytical relationships such as those presented in the paper have also been reported before [for example, see Chem. Mater. 28, 1304-1311 (2016)]. In fact, the machine learning analysis presented here appears to be highly inspired from the approach and work flow presented in Chem. Mater. 28, 1304-1311 (2016). Results and findings of the present work do not go any further from confirming the conventional wisdom. Specifically "What new insights have come out as a result of this work?" is not clear.

Response: We agree with the referee that the workflow of machine learning in this work is partly inspired from the paper of Chem. Mater. 28, 1304-1311 (2016), as cited in our manuscript. The machine learning in this work is mainly used to obtain an analytical expression of breakdown strength, allowing one to make quick predictions of breakdown strength for nanocomposites with available material parameters. Therefore, the focus of this part is to use a suitable and effective machine learning to connect with phase-field model rather than to design a new workflow or algorithm of machine learning. Furthermore, some specific improvements have been made in our machine learning, such as multiple rounds of screening and the specific interactions

between different fingerprints, as described in the section of *methods*. Therefore, the novelty of machine learning in this work is producing an analytical expression from the high-throughput phase-field simulation results using a simple but efficient approach, which can be practically very useful.

Furthermore, we also tried another machine learning method, back-propagation neural network (BPNN) to predict the breakdown strength of nanocomposites. More details are described in supporting information of section 5 on page 41-42. As shown in Fig. S9, the predictive ability of this neural network method is much stronger than the machine learning of LSR, with a higher coefficient of determination $R^2 = 0.983$. There is no doubt that both novelty and accuracy of BPNN are superior. However, as we have stated above, we want to obtain an analytical expression to help researchers make a quick estimation of breakdown strength for their material systems. To achieve this goal, the machine learning of LSR is a better choice than the neural network and other advanced methods. To clarify our purpose of machine learning in this work, some discussions are added in red text on page 13. “Aside from the LSR, we also tried another machine learning method, the back-propagation neural network (BPNN) with details described in the supporting information. In comparison to LSR, the BPNN exhibits better prediction ability of the breakdown strength. However, the BPNN cannot give an expression of breakdown strength as functions of the dielectric constant, electrical conductivity, and Young’s modulus, thereby it is less convenient for experimental researchers to make a quick estimation of the breakdown strength for a new material system.”

Based on the phase-field simulations and the machine learning, we found that the addition of oxides with lower dielectric constant and electrical conductivity such as Al_2O_3 , MgO , SiO_2 and TiO_2 in polymer matrix could lead to enhanced breakdown strength. Traditionally, researchers in this field preferred to fill high-dielectric-constant nanofillers into the polymer to improve the energy density by improving the effective dielectric constant. However, it is hard to significantly improve dielectric constant at a low volume fraction of the nanofillers. With the volume fraction of high-dielectric-constant nanofillers increasing, the breakdown strength may be severely decreased. As

a result, the energy density may be reduced due to the quadratic relationship between the energy density and the breakdown strength. Therefore, adding oxides with low dielectric constant and electrical conductivity into the polymer to enhance the breakdown strength at the expense of partially sacrificing the effective dielectric constant is one of our new insights. Moreover, this insight was verified by the successful synthesis and characterization of Al₂O₃/P(VDF-HFP) nanocomposites in this work. Therefore, the novelties of this work include phase-field model development, high-throughput simulations, machine learning, and experimental verification. In order to make our insights more clear, we have added some sentences **in red text** to describe our results, as follows:

The section of *abstract* on page 1 “It is found that the addition of oxides with lower dielectric constant and electrical conductivity such as Al₂O₃, MgO, SiO₂ and TiO₂ into the P(VDF-HFP) polymer can enhance the breakdown strength. ”

On pages 12-13 “In general, the Young’s modulus of the ceramic nanofillers is much larger than that of the polymer matrix, therefore seeking for nanofillers with lower dielectric constant and lower electrical conductivity will be more critical to improve the breakdown strength of nanocomposites.”

On page 16 “According to the analytical expression and the mechanism analysis above, if nanofillers with low dielectric constant and low electrical conductivity are added into the polymer, the breakdown strength of nanocomposites can be improved.”

On pages 16-17 “Specific examples include nanocomposites of P(VDF-HFP) filled with oxides such as Al₂O₃, SiO₂, MgO and TiO₂. The machine learning predicts that those nanocomposites should exhibit higher breakdown strength than pure polymer matrix, and this prediction is verified by both calculations and experiments in this work. ”

From the above reasons, the manuscript is more suitable for a specialized journal rather than Nature Communication, which emphasizes on novelty and impact.

Response: We are deeply thankful for the referee for her/his insights and comments, which are undoubtedly helpful to improve our manuscript. However, we respectfully

disagree with his/her recommendation for a more specialized journal. Here, we would like to briefly summarize our work again as follows:

- 1) We developed a comprehensive phase-field model of dielectric breakdown which simultaneously incorporates electrical, thermal, and mechanical effects.
- 2) We employed the developed model to perform high-throughput simulations and machine learning.
- 3) We produced an analytical expression of the breakdown strength as function of material parameters to quickly screen nanofillers.
- 4) We proposed to use oxides like Al_2O_3 , SiO_2 , MgO and TiO_2 as nanofillers to enhance the breakdown strength.
- 5) We performed targeted experiments to verify our simulation results.

We believe our results and conclusions will be helpful to understand the breakdown mechanisms and screen the nanofillers in future experiments. It will attract broad attention from the material and energy communities. Furthermore, we believe that the combination of high-throughput phase-field simulation, machine learning, and targeted experiments provide a paradigm to achieve the goal of Materials Genome Initiative project. The material system can also be extended to other functional composite materials such as thermoelectrics and solid electrolytes. Therefore, we hope we are able to convince the referee that the work is a significant advance and is of general interest, and the manuscript is suitable for publication in Nature Communications.

Reviewer #2

This manuscript presents a continuum, phase-field model for the breakdown of polymer dielectrics, which involves the electric energy, Joule heating, and strain. This model includes significantly more physics than the old Stark-Garton model and is able to reproduce the experimental data well to fairly well.

The authors also use machine-learning techniques on a set of ~400 simulations to arrive at an analytical expression that replicates well the results of the phase-field simulations, including data not included in the training set. The final part of the manuscript deals with predictions and experiments on nanocomposites.

The phase-field model represents a genuine advance over the Stark-Garton expression of breakdown and thus deserves publication in Nature Communications. However, the presentation and discussion in terms of figures and their descriptions left me wondering about the deeper physical issues and the expected accuracy, which have not been addressed fully. It is also hard to develop deeper physical insight from an examination of a few figures.

Response: We greatly appreciate the referee for her/his time to review our manuscript and give us above extremely encouraging comments. The enlightening suggestions are very helpful for improving this work. Thus, we have revised our manuscript accordingly, and the point-by-point responses to the referee's suggestions are enclosed as follows.

My suggestions are the following:

(i) The number of experimental data points used to compare to the results of phase-field modeling is rather limited, while the phase-field model uses parameters and approximations. The authors should discuss the extent to which their parameters are uniquely determined and the sensitivity of their results to parameter variations. This is particularly important because the physical properties of complex polymers vary between samples and are not known with high accuracy. What accuracy is expected? Are only the general trends expected to be reproduced or are the modeling results truly quantitative?

Response: We thank the referee for this important question and we totally agree that material parameters used in this simulation are particularly important.

1) In this work, three input parameters, the dielectric constant ϵ , electrical conductivity σ and Young's Modulus Y , of nanofillers and polymer matrix are all obtained from the existing experimental measurements, as shown in Table S6. However, as the referee mentioned, the physical properties are strongly dependent on the experimental environment and samples. Here, P(VDF-HFP) is used as the polymer matrix to fix the polymer material parameters, while the parameters of nanofillers are taken as variables to consider different nanofiller materials, as shown in Fig. 6(a). All material parameters

of the nanofiller are normalized using $\sigma_{\text{filler}}/\sigma_{\text{matrix}}$, $\varepsilon_{\text{filler}}/\varepsilon_{\text{matrix}}$, and $Y_{\text{filler}}/Y_{\text{matrix}}$. As we discussed in the main text on Page 12, “Moreover, the absolute value of the factor before the term $\ln(\sigma_{\text{filler}}/\sigma_{\text{matrix}})$ in the expression is larger than other factors before the other two terms, indicating that the breakdown strength of the P(VDF-HFP)-based nanocomposites is more sensitive to electrical conductivity than to dielectric constant and Young’s modulus, which is consistent with the calculations shown in Fig. 6(a)”. “More specifically, with the electrical conductivity, the most sensitive material parameter, increasing from 2.12×10^{-13} S/m to 2.12×10^{-7} S/m while maintaining the dielectric constant at 13.5 and the Young’s modulus at 0.982 GPa, the normalized breakdown strength decreases from about 1.34 and 0.32. However, with the Young’s modulus, the least sensitive material parameter, increasing from 0.982 MPa to 982 GPa while maintaining the dielectric constant at 13.5 and the electrical conductivity at 2.12×10^{-10} S/cm, the normalized breakdown strength just increases from 0.42 to 1.04.”

2) On the other hand, the breakdown strengths measured in experiments show the Weibull distribution due to numerous and complex factors. Because of the uncontrollable accuracies of input physical properties and output breakdown strength, the analytical expression from machine learning is expected to be semiquantitative for predicting the breakdown strength and screening nanofillers in nanocomposites. The accuracy of the calculated breakdown strength partly depends on the input parameters of materials. For example, for BaTiO₃ and Al₂O₃, the variations of their physical parameters caused by experiment conditions is less matchable to the large intrinsic differences of electric and electrical properties, only leading to small fluctuations. Thus, it doesn’t affect our conclusions that the introduction of oxides such as Al₂O₃, SiO₂, MgO and TiO₂ with lower dielectric constant and lower electrical conductivity can enhance the breakdown strength of the polymer.

In order to make our conclusions more clear, we have revised some red texts on page 1 and page 11.

On page 1 “It can be used to semiquantitatively predict the breakdown strength of the P(VDF-HFP)-based nanocomposites with a wide variety of candidate nanofillers.”

On page 11 “More specifically, with the electrical conductivity, the most sensitive material parameter, increasing from 2.12×10^{-13} S/m to 2.12×10^{-7} S/m while maintaining the dielectric constant at 13.5 and the Young’s modulus at 0.982GPa, the normalized breakdown strength decreases from about 1.34 to 0.32. However, with the Young’s modulus, the least sensitive material parameter, increasing from 0.982 MPa to 982 GPa while maintaining the dielectric constant at 13.5 and the electrical conductivity at 2.12×10^{-10} S/cm, the normalized breakdown strength just increases from 0.42 to 1.04.”

(ii) I would like to see a deeper physical description of the model in the main text, perhaps instead of some figures if space is a problem.

Response: We thank the reviewer for this enlightening suggestion. We added a schematic diagram on phenomenological energy profile of the breakdown process in Fig. 4(a). In the main text of the revised manuscript, we have added following texts in red on page 9:

“Fig. 4a phenomenologically shows the variation of the energy profile for polymers under physical stimuli. Curve 1 describes a double-well energy density as function of the order parameter η , and the energy barrier height between $\eta=0$ (unbroken phase) and $\eta=1$ (broken phase) represents how difficult a local point can be broken down. With the increase of external physical stimuli such as the electric field E or temperature T or both, the energy barrier drops and energy profile tilts, leading to a lower energy state at $\eta=1$ than that at $\eta=0$, as illustrated by the variation from curve 1 to curve 3 shown in Fig. 4(a). Once the physical stimuli is sufficiently large, the energy barrier vanishes and the breakdown occurs. The height of this energy barrier is also related to the material parameters including the dielectric constant ϵ , the electrical conductivity σ and the Young’s Modulus Y . For a material with higher dielectric constant ϵ , higher electrical conductivity σ and lower Young’s Modulus Y , the energy barrier is lower. Thus, the breakdown strength will be lower. ”

More captions are added for Fig.4 on page 47.

Fig. 4 Schematic diagrams for understanding and classifying breakdown mechanisms. (a) Variation of the energy profile as function of the order parameter for polymers under physical stimuli. (b) Identification of breakdown mechanisms in polymer-based material systems. Examples of PVDF, P(VDF-TrFE-CTFE), PANI, C-BCB/BNNS, and BTNFS/PI are from experiments²³⁻²⁷.

(iii) Clearly, the phase-field model uses a continuum description and ignores atomic-scale effects, such as defects and material inhomogeneities. A discussion of these left-out effects would benefit the readers.

Response: We thank the reviewer for such a constructive suggestion. Yes, it is important to discuss the defects effects for benefiting the readers. We have added brief discussions on page 14 of the revised manuscript:

“Many factors such as voids, space charge effects⁴⁰⁻⁴² and incomplete crystallization^{11,13,43,44} of the polymer that are not incorporated into the phase-field model may cause this difference between the phase-field-based machine learning and the experimental results. For example, the existence of voids may cause partial discharge at the void/polymer interfaces where the local electric field is intensified. A high concentration of space charges can cause the increases in the electrical

conductivity. The crystallinity of polymers may also affect the breakdown strength, arising from the different transport behaviors of charge carriers in the amorphous phase and crystalline phase. Unfortunately, the introduction of a large number of nanofillers can easily introduce these effects due to the incompatibility of ceramics nanofillers and polymers. Therefore, the preparation of high-quality polymer nanocomposites is extremely important to achieve a high breakdown strength.”

The referee has provided us very constructive comments, which are helpful to us for improving the readability of our manuscript. We therefore sincerely thank the referee again for her/his encouragement and suggestions.

Reviewer #3

This manuscript developed a comprehensive phase-field model to investigate electric-thermal-mechanical breakdown of polymer-based dielectrics. Until now, both the theoretical and experimental study of the mechanism of breakdown strength of polymer-based composites are rather insufficient to obtain a definite conclusion compared with other dielectric properties. Due to the various influencing factors, such as the electrode system, measuring environment, the shape of the sample, the intrinsic properties of materials and so on, it is unrealistic to depict the breakdown phenomenon with a unified theory. However, it does not affect the contribution of this paper revealing the internal breakdown mechanism of polymer-based composites. Comparison between the calculation and the experiment results indicates that the incorporation of the phase field method is effective. It is of great significant to guide the experiment design with the conclusions from the phase field calculation and the linear regression.

Based on different energies, this model could identify the breakdown mechanisms and predict the breakdown strength of polymer-based dielectrics under different stimulus. Then, high-throughput calculations and machine learning were conducted to produce an analytical expression of breakdown strength. I think this work is interesting and very helpful to design and screen nanocomposites for experiments. I recommend this manuscript to be published after the following comments are addressed.

Response: We greatly appreciate the referee’s highly encouraging comments on our work. Yes, we fully agree with the referee that the dielectric breakdown is rather complicated phenomenon and hardly depicted with a unified theory by considering all factors. This work is aimed at developing a relatively comprehensive model to help us understand the internal breakdown process and provide some theoretical guidance to experiments.

1. The author mentioned “assuming that the initial breakdown phase is nucleated from the two needle electrodes”. Why does the breakdown path start from the electrodes? Please give enough explanation.

Response: We thank the referee for this question. This assumption is based on two considerations. 1) Due to the huge property contrast between the polymer dielectrics and the metal electrode, the electric field may concentrate at the metal/polymer interface, making this area vulnerable. Therefore, we assume that the initial breakdown phase is nucleated from this area. We have simply simulated the local electric field distribution around a Cu needle electrode deposited at P(VDF-HFP) polymer. As shown in Fig. R1, the electric field at the electrode/polymer interface is much higher than that at other region, rationalizing this assumption.

Fig.R1 The local electric field distribution of a Cu electrode and P(VDF-HFP) dielectric.

2) When operating at high voltage and high temperature, charge injection from electrodes into the dielectric may make the area around electrodes hot spots, thereby

triggering the nucleation of breakdown at this area. This has been demonstrated by some experiments and simulations (for example, see *Advanced Materials*, 2017, 29(35): 1701864. and *Materials Letters*, 2015, 141: 14-19.).

Therefore, based on above reasons, we assume the initial breakdown phase is nucleated from the two needle electrodes.

2. In this work, the matrix polymer is P(VDF-HFP). Does this model also work in other systems? How about the microstructure of nanocomposites? Does it consider the state of the molecular chain or crystallinity of polymer?

Response: We thank the reviewer for these great questions. Yes, this comprehensive breakdown phase-field model works for other systems. As long as the necessary materials parameters of the polymer and filler are available, this model can be used to simulate the breakdown process and predict the breakdown strength with certain assumptions. In this work, P(VDF-HFP) polymer and oxide nanoparticles are taken as the example to validate the model. With regard to microstructure, this model should work for any microstructure of the nanocomposite. As shown in our previous publication (*Adv. Mater.* 30, 1704380 (2018)), the microstructure effect on the electrostatic breakdown has been systematically investigated. However, the focus of this work is not on the microstructure effect but the filler material effect. For effects of the molecular chain and the crystallinity of polymers, we didn't include them in this model currently, because we assume the polymer phase a homogeneous phase. If distinguishing the crystalline phase and amorphous phase and taking them as two different phases in the polymer, this model can also be used to simulate the crystallinity effect on the breakdown behavior in pure polymer dielectric. This question gives us a next-step direction to expand this model to explore more interesting research topics, e.g., the microstructure of the pure polymer on the breakdown property. We thank the reviewer again for so nice questions.

3. For dielectric breakdown, there are so many factors affecting this process. Sometimes, the extrinsic factors, e.g., defects, impurities or air hole, may also be important.

Response: We thank the referee for this great comment. Yes, we totally agree that the breakdown process may be affected by extrinsic factors such as the pinhole, defects and void. All these factors may cause the concentration of local electric field due to the large contrast of dielectric or electrical properties. The region where these defects are located can easily become the hot spot to trigger the sequential breakdown process before the intrinsic electric breakdown occurs. However, in this work, the simulation of breakdown process is performed under an ideal condition to investigate the intrinsic breakdown mechanism, without considering those extrinsic factors considering that they are hard to control in experiments. By incorporating those defects into the microstructure, their effects on breakdown can actually be studied. If necessary, we can design some simulations to investigate effects of those extrinsic factors on the breakdown behavior in the future. We thank the reviewer again for this nice comment.

4. Using the analytical expression of dielectric breakdown by machine learning, it has found that fillers like Al_2O_3 or MgO can improve the breakdown strength of nanocomposites. How can we use the expression to find more new materials?

Response: This is a great question, and we thank the reviewer very much. The analytical expression can be employed to readily predict the breakdown strength of polymer nanocomposites filled with nanoparticles with available material parameters. Thus, if the material parameters, including the dielectric constant, electrical conductivity, and Young's modulus of a new material (e.g., AB) are available, we could quickly predict the breakdown strength using equation (2). If the predicted breakdown strength is greater than 1.0, this new material AB can be added into the polymer to enhance the breakdown strength. The key is to know these material parameters of AB.

5. The introduction of fillers like Al_2O_3 or MgO can improve the breakdown strength of nanocomposites. But those fillers also lead to the decrease of dielectric constant due to the low intrinsic dielectric constant. So if we want to get high energy density of nanocomposites, how to balance these two parameters?

Response: We thank the referee for pointing this important question out, which have

always been intensively investigated by researchers by manipulating these two key parameters according to the linear approximation $U = \frac{1}{2} \epsilon_0 \epsilon_r E_b^2$. However, the inverted coupling relationship between the dielectric constant and breakdown strength determines that a pure material can hardly possess a high dielectric constant and a high breakdown strength. As an alternative approach, polymer nanocomposites have been proposed and explored for years to improve the energy density by taking advantages of the high breakdown strength of polymer matrix and the high dielectric constant of ceramics. However, the electric field may concentrate near the interfaces in nanocomposites due to large mismatch of nanofillers and matrix, leading to the decrease of breakdown strength. Thus, even in nanocomposites, it is also hard to keep the dielectric constant and breakdown strength at high levels concurrently. To date, one effective method demonstrated by simulations and experiments (such as Adv. Mater., 2018, 30(2): 1704380, Adv. Funct. Mater., 2017, 27(20): 1606292. and Adv. Mater., 2017, 29(35): 1701864.) is building multilayer structures. For example, we can prepare one layer of high voltage resistance and one layer of high polarization by adding different nanofillers and make them into sandwich structure to improve the energy density.

6. Whether the simulation model is 2D or 3D. if it is 2D, when extending to a three-dimensional condition, the circle representing the ceramic particles will become cylinder rather sphere. It will deviate from the actual situations and the well agreement between the calculation and the experiment will be less convincing.

Response: We thank the referee for raising this important issue. We totally agree that there are some inevitable differences when extending 2D simulation to 3D simulation. To clarify this question, we discuss several points as follows:

1) In this model, the local energy density is considered as the breakdown criterion. If the energy density at one point exceeds the corresponding critical energy density, the breakdown path will grow. Thus, the local electric field distribution is one of the most important factors, because it is strongly related with the energy density of every point.

In order to compare different conditions, we simulate the electric field distributions of three states of BTO/(PVDF-HFP): 2D circle, 3D sphere, and 3D cylinder, as shown in Fig. R2. As the cross sections shown in Fig. R2(c) and R2(e), the local electric field distributions have similar distributions when 2D circle extends to 3D cylinder and 3D sphere when compared with 2D circle. However, 3D sphere can cause more severe concentration of local electric field along applied electric field. When considering the 3D distributions as shown in Fig. R2(b) and R2(d), the distribution is more dependent on the shape of nanofillers.

Fig. R2 The local electric field distributions of (a) 2D circle, (b) and (c) 3D cylinder, (d) and (e) 3D sphere.

2) In order to verify our analysis above, we perform a set of 2D and 3D simulations to compare their results. In the simulation, 10 random microstructures for 5% $\text{Al}_2\text{O}_3/\text{P}(\text{VDF-HFP})$ nanocomposites are generated to calculate the corresponding breakdown strength, as shown in Fig. R3. It can be seen that the 3D simulation gives an average value of 1.45 for $E_b^{\text{composite}} / E_b^{\text{matrix}}$, which is slightly larger than the average value of 1.42 from 2D simulation. Although there exist small deviations due to the change of dimensionality, it doesn't affect the capability of semiquantitatively predicting the breakdown strength by phase-field model. It doesn't affect the conclusion

that the addition of oxides with lower dielectric constant and electrical conductivity such as Al_2O_3 , MgO , SiO_2 and TiO_2 into the P(VDF-HFP) polymer can enhance the breakdown strength neither. Therefore, we used 2D high-throughput simulations in this work for reducing the huge computational cost. Of course, if the computational resource is allowed, 3D simulation is definitely the best choice, particularly when modeling polymer nanocomposites filled with nanofibers and nanosheets,

Fig. R3 The evolution of breakdown path obtained in (a)-(c) 3D and (d)-(f) 2D simulations for 5% Al_2O_3 /P(VDF-HFP) nanocomposites. 10 random microstructures are generated and used for each set of simulation.

7. When simulating the thermal breakdown, whether the heat dissipation is considered? In other works, whether the temperature (363K) is only used to change the temperature-dependent electrical conductivity or used as the ambient temperature?

Response: We thank the reviewer for this question. In this simulation, the heat dissipation is not considered. As the thickness of nanocomposite films in experiments is only about $\sim 10 \mu\text{m}$, thus the heat can immediately dissipate without causing the increase in temperature according to our thermal steady-state simulation. Here, we exhibit the example of thermal simulation results of two dielectric films when operating at 363K and 200kV/mm by solving $\kappa \nabla T(\mathbf{r}, t) + \sigma(\mathbf{r}, t) E^2 = 0$ and $\mathbf{n}(-\kappa \nabla T) = h(T - T_{\text{amb}})$.

The electrical conductivity is set at 10^{-11} S/m and the convective heat transfer coefficient is $10 \text{ Wm}^{-2} \text{ K}^{-1}$. As shown in Fig. R4, the temperature in film (a) is almost equal to the ambient temperature. However, when the thickness increases to 10mm, the temperature can reach about 560K.

Fig.R4 The steady-state temperature distributions in dielectric films with different sizes (a) $1\mu\text{m}\times 10\text{mm}\times 10\text{mm}$ and (b) $10\text{mm}\times 10\text{mm}\times 10\text{mm}$.

Therefore, in this simulation of thin polymer film, the internal temperature can be regarded as uniform and same as the ambient temperature.

8. Supporting Information [line601-604]: The kinetic coefficient L_0 is not given.

Response: We thank the reviewer for this kind reminder. Here, the kinetic coefficient L_0 is related to the breakdown phase wall mobility and is assigned a value of $1.0\text{ m}^2\text{s}^{-1}\text{N}^{-1}$ due to the lack of experimental data. We have added it into the manuscript on page 26.

9. Supporting Information [equation 8-9]: Can you give a detailed derivation for eq.8 to eq.9.

Response: We thank the reviewer for this question. Yes, the derivation from Eq.(8) to Eq.(9) is as following:

$$\begin{aligned} \frac{\eta(\mathbf{r}, t + \Delta t) - \eta(\mathbf{r}, t)}{\Delta t} = & -L_0 H(f_{\text{elec}} + f_{\text{Joule}} + f_{\text{strain}} - f_{\text{critical}}) \left[\begin{array}{c} \frac{\partial f_{\text{sep}}(\eta)}{\partial \eta(\mathbf{r}, t)} + \frac{\partial f_{\text{elec}}}{\partial \eta(\mathbf{r}, t)} \\ + \frac{\partial f_{\text{Joule}}}{\partial \eta(\mathbf{r}, t)} + \frac{\partial f_{\text{strain}}}{\partial \eta(\mathbf{r}, t)} \end{array} \right] \\ & + L_0 \left(H(f_{\text{elec}} + f_{\text{Joule}} + f_{\text{strain}} - f_{\text{critical}}) - \frac{1}{2} \right) \gamma \nabla^2 \eta(\mathbf{r}, t) + \frac{1}{2} L_0 \gamma \nabla^2 \eta(\mathbf{r}, t + \Delta t). \end{aligned} \quad (8)$$

$$\begin{aligned}
&\Rightarrow \\
&\eta(\mathbf{r}, t + \Delta t) - \eta(\mathbf{r}, t) \\
&= -L_0 \Delta t H(f_{\text{elec}} + f_{\text{Joule}} + f_{\text{strain}} - f_{\text{critical}}) \left[\frac{\partial f_{\text{sep}}(\eta)}{\partial \eta(\mathbf{r}, t)} + \frac{\partial f_{\text{elec}}}{\partial \eta(\mathbf{r}, t)} + \frac{\partial f_{\text{Joule}}}{\partial \eta(\mathbf{r}, t)} + \frac{\partial f_{\text{strain}}}{\partial \eta(\mathbf{r}, t)} \right] \\
&+ L_0 \Delta t \left(H(f_{\text{elec}} + f_{\text{Joule}} + f_{\text{strain}} - f_{\text{critical}}) - \frac{1}{2} \right) \gamma \nabla^2 \eta(\mathbf{r}, t) + \frac{1}{2} L_0 \Delta t \gamma \nabla^2 \eta(\mathbf{r}, t + \Delta t).
\end{aligned} \tag{8.1}$$

$$\begin{aligned}
&\Rightarrow \\
&\eta(\mathbf{r}, t + \Delta t) - \eta(\mathbf{r}, t) \\
&= -L_0 \Delta t H(f_{\text{elec}} + f_{\text{Joule}} + f_{\text{strain}} - f_{\text{critical}}) \left[\frac{\partial f_{\text{sep}}(\eta)}{\partial \eta(\mathbf{r}, t)} + \frac{\partial f_{\text{elec}}}{\partial \eta(\mathbf{r}, t)} + \frac{\partial f_{\text{Joule}}}{\partial \eta(\mathbf{r}, t)} + \frac{\partial f_{\text{strain}}}{\partial \eta(\mathbf{r}, t)} \right] \\
&+ L_0 \Delta t H(f_{\text{elec}} + f_{\text{Joule}} + f_{\text{strain}} - f_{\text{critical}}) \gamma \nabla^2 \eta(\mathbf{r}, t) - \frac{1}{2} L_0 \Delta t \gamma \nabla^2 \eta(\mathbf{r}, t) + \frac{1}{2} L_0 \Delta t \gamma \nabla^2 \eta(\mathbf{r}, t + \Delta t).
\end{aligned} \tag{8.2}$$

$$\begin{aligned}
&\Rightarrow \\
&\eta(\mathbf{r}, t + \Delta t) - \eta(\mathbf{r}, t) \\
&= -L_0 \Delta t H(f_{\text{elec}} + f_{\text{Joule}} + f_{\text{strain}} - f_{\text{critical}}) \left[\frac{\partial f_{\text{sep}}(\eta)}{\partial \eta(\mathbf{r}, t)} + \frac{\partial f_{\text{elec}}}{\partial \eta(\mathbf{r}, t)} + \frac{\partial f_{\text{Joule}}}{\partial \eta(\mathbf{r}, t)} + \frac{\partial f_{\text{strain}}}{\partial \eta(\mathbf{r}, t)} - \gamma \nabla^2 \eta(\mathbf{r}, t) \right] \\
&- \frac{1}{2} L_0 \Delta t \gamma \nabla^2 \eta(\mathbf{r}, t) + \frac{1}{2} L_0 \Delta t \gamma \nabla^2 \eta(\mathbf{r}, t + \Delta t).
\end{aligned} \tag{8.3}$$

Performing Fourier transformation on both sides of Eq.(8.3), we have

$$\begin{aligned}
&\tilde{\eta}(\mathbf{q}, t + \Delta t) - \tilde{\eta}(\mathbf{q}, t) \\
&= -L_0 \Delta t \left\{ H(f_{\text{elec}} + f_{\text{Joule}} + f_{\text{strain}} - f_{\text{critical}}) \left[\frac{\partial f_{\text{sep}}(\eta)}{\partial \eta(\mathbf{r}, t)} + \frac{\partial f_{\text{elec}}}{\partial \eta(\mathbf{r}, t)} + \frac{\partial f_{\text{Joule}}}{\partial \eta(\mathbf{r}, t)} + \frac{\partial f_{\text{strain}}}{\partial \eta(\mathbf{r}, t)} - \gamma \nabla^2 \eta(\mathbf{r}, t) \right] \right\}_{\mathbf{q}} \\
&- \frac{1}{2} L_0 \Delta t \gamma (i\mathbf{q} \cdot i\mathbf{q} \tilde{\eta}(\mathbf{q}, t)) + \frac{1}{2} L_0 \Delta t \gamma (i\mathbf{q} \cdot i\mathbf{q} \tilde{\eta}(\mathbf{q}, t + \Delta t)).
\end{aligned} \tag{8.4}$$

$$\begin{aligned}
&\Rightarrow \\
&\tilde{\eta}(\mathbf{q}, t + \Delta t) - \tilde{\eta}(\mathbf{q}, t) \\
&= -L_0 \Delta t \left\{ H(f_{\text{elec}} + f_{\text{Joule}} + f_{\text{strain}} - f_{\text{critical}}) \left[\frac{\partial f_{\text{sep}}(\eta)}{\partial \eta(\mathbf{r}, t)} + \frac{\partial f_{\text{elec}}}{\partial \eta(\mathbf{r}, t)} + \frac{\partial f_{\text{Joule}}}{\partial \eta(\mathbf{r}, t)} + \frac{\partial f_{\text{strain}}}{\partial \eta(\mathbf{r}, t)} - \gamma \nabla^2 \eta(\mathbf{r}, t) \right] \right\}_{\mathbf{q}} \\
&+ \frac{1}{2} L_0 \Delta t \gamma q^2 \tilde{\eta}(\mathbf{q}, t) - \frac{1}{2} L_0 \Delta t \gamma q^2 \tilde{\eta}(\mathbf{q}, t + \Delta t).
\end{aligned} \tag{8.5}$$

$$\begin{aligned}
&\Rightarrow \\
&\tilde{\eta}(\mathbf{q}, t + \Delta t) + \frac{1}{2} L_0 \Delta t \gamma q^2 \tilde{\eta}(\mathbf{q}, t + \Delta t) \\
&= \tilde{\eta}(\mathbf{q}, t) + \frac{1}{2} L_0 \Delta t \gamma q^2 \tilde{\eta}(\mathbf{q}, t) \\
&- L_0 \Delta t \left\{ H(f_{\text{elec}} + f_{\text{Joule}} + f_{\text{strain}} - f_{\text{critical}}) \left[\frac{\partial f_{\text{sep}}(\eta)}{\partial \eta(\mathbf{r}, t)} + \frac{\partial f_{\text{elec}}}{\partial \eta(\mathbf{r}, t)} + \frac{\partial f_{\text{Joule}}}{\partial \eta(\mathbf{r}, t)} + \frac{\partial f_{\text{strain}}}{\partial \eta(\mathbf{r}, t)} - \gamma \nabla^2 \eta(\mathbf{r}, t) \right] \right\}_{\mathbf{q}}
\end{aligned} \tag{8.6}$$

$$\begin{aligned}
&\Rightarrow \\
&\left(1 + \frac{1}{2} L_0 \Delta t \gamma q^2 \right) \tilde{\eta}(\mathbf{q}, t + \Delta t) \\
&= \left(1 + \frac{1}{2} L_0 \Delta t \gamma q^2 \right) \tilde{\eta}(\mathbf{q}, t) \\
&- L_0 \Delta t \left\{ H(f_{\text{elec}} + f_{\text{Joule}} + f_{\text{strain}} - f_{\text{critical}}) \left[\frac{\partial f_{\text{sep}}(\eta)}{\partial \eta(\mathbf{r}, t)} + \frac{\partial f_{\text{elec}}}{\partial \eta(\mathbf{r}, t)} + \frac{\partial f_{\text{Joule}}}{\partial \eta(\mathbf{r}, t)} + \frac{\partial f_{\text{strain}}}{\partial \eta(\mathbf{r}, t)} - \gamma \nabla^2 \eta(\mathbf{r}, t) \right] \right\}_{\mathbf{q}}
\end{aligned} \tag{8.7}$$

$$\begin{aligned}
&\Rightarrow \\
&\tilde{\eta}(\mathbf{q}, t + \Delta t) \\
&= \tilde{\eta}(\mathbf{q}, t) \\
&- \frac{L_0 \Delta t}{\left(1 + \frac{1}{2} L_0 \Delta t \gamma q^2 \right)} \left\{ H(f_{\text{elec}} + f_{\text{Joule}} + f_{\text{strain}} - f_{\text{critical}}) \left[\frac{\partial f_{\text{sep}}(\eta)}{\partial \eta(\mathbf{r}, t)} + \frac{\partial f_{\text{elec}}}{\partial \eta(\mathbf{r}, t)} + \frac{\partial f_{\text{Joule}}}{\partial \eta(\mathbf{r}, t)} + \frac{\partial f_{\text{strain}}}{\partial \eta(\mathbf{r}, t)} - \gamma \nabla^2 \eta(\mathbf{r}, t) \right] \right\}_{\mathbf{q}}
\end{aligned} \tag{8.8}$$

In Eq.(8.8), $\nabla^2 \eta(\mathbf{r}, t)$ can be calculated in Fourier space at first and then transformed back to the real space, i.e.,

$$\gamma \nabla^2 \eta(\mathbf{r}, t) = \left\{ -\gamma q^2 \tilde{\eta}(\mathbf{q}, t) \right\}_{\mathbf{r}} \tag{8.9}$$

Substituting Eq.(8.9) into Eq.(8.8), we can get Eq.(9)

$$\begin{aligned} \tilde{\eta}(\mathbf{q}, t + \Delta t) = & \tilde{\eta}(\mathbf{q}, t) + \frac{L_0 \Delta t}{(1 + 0.5 L_0 \gamma \Delta t q^2)} \left\{ -H(f_{\text{elec}} + f_{\text{Joule}} + f_{\text{strain}} - f_{\text{critical}}) \times \right. \\ & \left. \left[\frac{\partial f_{\text{sep}}(\eta)}{\partial \eta(\mathbf{r}, t)} + \left\{ \gamma q^2 \tilde{\eta}(\mathbf{q}, t) \right\}_{\mathbf{r}} + \frac{\partial f_{\text{elec}}}{\partial \eta(\mathbf{r}, t)} + \frac{\partial f_{\text{Joule}}}{\partial \eta(\mathbf{r}, t)} + \frac{\partial f_{\text{strain}}}{\partial \eta(\mathbf{r}, t)} \right] \right\}_{\mathbf{q}} \end{aligned} \quad (9)$$

10. Supporting Information [equation 11] Please explain the equation more detailed. Where does the expression $(\eta^3 (10 - 15\eta + 6\eta^2))$ come from?

Response: We thank the reviewer for this question. We rewrite Eq.(11) here

$$\begin{aligned} \varepsilon_{ij}(\mathbf{r}) = & \eta^3 (10 - 15\eta + 6\eta^2) \varepsilon_{ij}^{\text{B}} + \left[1 - \eta^3 (10 - 15\eta + 6\eta^2) \right] \times \\ & \left\{ \rho^3 (10 - 15\rho + 6\rho^2) \varepsilon_{ij}^{\text{F}} + \left[1 - \rho^3 (10 - 15\rho + 6\rho^2) \right] \varepsilon_{ij}^{\text{M}} \right\}. \end{aligned} \quad (11)$$

Eq.(11) describes the spatial-dependent dielectric constant determined by the microstructure. The term $\eta^3 (10 - 15\eta + 6\eta^2)$ is function which can describes a diffuse interface between $\eta=0$ and $\eta=1$. As shown in Fig. R4, when $\eta=0$, $\eta^3 (10 - 15\eta + 6\eta^2) = 0$, and when $\eta=1$, $\eta^3 (10 - 15\eta + 6\eta^2) = 1$. When $0 < \eta < 1$, this function gives a diffuse change from 0 to 1.

In the breakdown phase, i.e., $\eta=1$, Eq.(11) becomes $\varepsilon_{ij}(\eta(\mathbf{r})=1) = \varepsilon_{ij}^{\text{B}}$. In the unbroken phase, if a local point belongs to the polymer matrix phase, i.e., $\rho=0$, Eq.(11) becomes $\varepsilon_{ij}(\eta(\mathbf{r})=1, \rho(\mathbf{r})=0) = \varepsilon_{ij}^{\text{M}}$. If a local point in the unbroken phase belongs to the filler phase, i.e., $\rho=1$, Eq.(11) becomes $\varepsilon_{ij}(\eta(\mathbf{r})=1, \rho(\mathbf{r})=1) = \varepsilon_{ij}^{\text{F}}$.

Fig. R4. The diffuse interface function $\eta^3(10-15\eta+6\eta^2)$.

In the end, we thank the reviewer again for so above nice comments and questions.

REVIEWERS' COMMENTS:

Reviewer #1 (Remarks to the Author):

In the revised version of the manuscript the authors were able to address my comments satisfactorily. I after going through the response letter and revised paper, I am happy to recommend the paper for publication in Nat. Comm.

Reviewer #2 (Remarks to the Author):

The authors have provided very comprehensive responses to the reviewers' comments. I have no further concerns and recommend this manuscript for publication in Nature Communications.

Reviewer #3 (Remarks to the Author):

The paper has been carefully revised and the response is exhaustive. I am really interested in the incorporation of phase field to predict the breakdown strength of the composites system and I think it will bring a new broad of perspective on dielectric study. However, I have verified the derivation of phase field and find a serious confusion about the equation (12) in supporting information.

The equation (12) is used to solve the derivative of f_{ele} with respect to phase variable η and can be obtained only provided that the electric field E is independent with phase variable η . However, as we known, the distribution of breakdown phase (represented as the variation of phase variable η) can strong affect the variation of electric field E . Therefore, E can not be reckoned as a constant when solving the derivative of f_{ele} with respect to phase variable.

I would like to recommend this paper to be accepted after addressing the above problem

Response to the Reviewers' comments:

Reviewer #1

In the revised version of the manuscript the authors were able to address my comments satisfactorily. I after going through the response letter and revised paper, I am happy to recommend the paper for publication in Nat. Comm.

Response: We are deeply thankful for the referee to review our manuscript and the agreement for the publication.

Reviewer #2

The authors have provided very comprehensive responses to the reviewers' comments. I have no further concerns and recommend this manuscript for publication in Nature Communications.

Response: We really appreciate the referee for the recommendation.

Reviewer #3

The paper has been carefully revised and the response is exhaustive. I am really interested in the incorporation of phase field to predict the breakdown strength of the composites system and I think it will bring a new broad of perspective on dielectric study. However, I have verified the derivation of phase field and find a serious confusion about the equation (12) in supporting information.

The equation (12) is used to solve the derivative of f_{ele} with respect to phase variable η and can be obtained only provided that the electric field E is independent with phase variable η . However, as we known, the distribution of breakdown phase (represented as the variation of phase variable η) can strong affect the variation of electric field E . Therefore, E can not be reckoned as a constant when solving the derivative of f_{ele} with respect to phase variable.

I would like to recommend this paper to be accepted after addressing the above problem

Response: We thank the referee for the positive evaluation on our work. Yes, the electric field E can strongly affect the breakdown and it is related to the distribution of the phase-field variable η . However, E is only implicitly linked to η and the dependence of E on η has been considered through the continuity equation (17). Therefore, when performing the derivative of f_{ele} with respect to η , $dE/d\eta$ should be regarded as zero. This is similar to other cases such as ferroelectric and ferromagnetic systems. For example, in the phase-field model of ferroelectrics, the order parameter is polarization P , and there is one term $P \cdot E$ in the electric energy. When performing first-order derivative of the electric energy with respect to P , the electric field E is taken explicitly independent on P although P must affect the distribution of E in principle. In fact, the effect of P on E has already been taken into account through the Poisson equation: $\epsilon_0 \epsilon_r \nabla \cdot E = -\nabla \cdot P$. Therefore, for Eq.(12) in the supplementary information, the electric field E is not explicitly dependent on η , and $dE/d\eta$ should be zero. Hopefully this explanation is understandable to the reviewer.